# Screening the Protective Agents Able to Improve the Survival of Lactic Acid Bacteria Strains Subjected to Spray Drying Using Several Key Enzymes Responsible for Carbohydrate Utilization

**DOI:** 10.3390/microorganisms12061094

**Published:** 2024-05-28

**Authors:** Jing Liu, Shanshan Xie, Mengfan Xu, Xiaoying Jiang, Qian Wang, Hongfei Zhao, Bolin Zhang

**Affiliations:** Beijing Key Laboratory of Forest Food Processing and Safety, College of Biological Science and Biotechnology, Beijing Forestry University, Beijing 100083, China; liujing888y@126.com (J.L.); xiess0907@126.com (S.X.); xumengfan1234@163.com (M.X.); jiangxiaoying0730@163.com (X.J.); wang18741875882@163.com (Q.W.); zhaohf518@163.com (H.Z.)

**Keywords:** spray drying, lactic acid bacteria strains (LAB strains), enzyme activity, carbohydrate utilization, recovered skim milk (RSM)

## Abstract

The aim of this study was to identify the most effective protectants for enhancing the viability of specific lactic acid bacteria (LAB) strains (*Lactobacillus delbrueckii* subsp. *bulgaricus* CICC 6097, *Lactiplantibacillus plantarum* CICC 21839, *Lactobacillus acidophilus* NCFM) by assessing their enzymatic activity when exposed to spray drying (inlet/outlet temperature: 135 °C/90 °C). Firstly, it was found that the live cell counts of the selected LAB cells from the 10% (*w*/*v*) recovered skim milk (RSM) group remained above 10^7^ CFU/g after spray drying. Among all the three groups (1% *w*/*v* RSM group, 10% *w*/*v* RSM group, and control group), the two enzymes pyruvate kinase (PK) and lactate dehydrogenase (LDH) were more sensitive to spray drying than hexokinase (HK) and β-galactosidase (β-GAL). Next, transcriptome data of *Lb. acidophilus* NCFM showed that 10% (*w*/*v*) RSM improved the down-regulated expressions of genes encoding PK (*pyk*) and LDH (*ldh*) after spray drying compared to 1% (*w*/*v*) RSM. Finally, four composite protectants were created, each consisting of 10% (*w*/*v*) RSM plus a different additive—sodium glutamate (CP-A group), sucrose (CP-B group), trehalose (CP-C group), or a combination of sodium glutamate, sucrose, and trehalose (CP-D group)—to encapsulate *Lb. acidophilus* NCFM. It was observed that the viable counts of strain NCFM (8.56 log CFU/g) and enzymatic activity of PK and LDH in the CP-D group were best preserved compared to the other three groups. Therefore, our study suggested that measuring the LDH and PK activity could be used as a promising tool to screen the effective spray-dried protective agent for LAB cells.

## 1. Introduction

The health benefits associated with lactic acid bacteria (LAB), such as mitigating hypertension, preventing cancer, and alleviating allergic symptoms, have been widely reported [1]. The International Dairy Federation recommends that the desirable probiotic products conferring beneficial effects on the host should contain at least the live counts of LAB that are more than 10^6^ CFU/g [2]. Consequently, the development of a dry probiotic powder that can consistently generating a substantial number of live cells is an essential prerequisite for preserving the efficacy of probiotics during processing [3]. Commonly, spray drying is a cost-effective method that produces active probiotic powder compared to freeze drying [4]. Nonetheless, during spray drying, the rapid dehydration of atomized droplets in hot air imposes diverse stresses upon LAB cells, including heat and dehydration stress, which may compromise cell and essential enzymes, thereby diminishing cell viability [5]. Thus, to protect LAB from these stresses during spray drying, the incorporation of a protective agent is the most effective approach. Recovered skim milk (RSM) is frequently employed as a protective agent due to its abundant nutritional constituents, comprising casein, whey protein, lactose, and inorganic salts [3]. Numerous studies have shown that elevating the concentration of protectants helps enhance the resistance of cells to heat [6]. For instance, a study conducted by Suo et al. (2021) showed that the survival of *Lacticaseibacillus casei* BL23 increased from 0.71% to 26.5% with an increase in RSM concentration from 5% to 30%. This enhancement in survival rate was ascribed to the potentiated cellular stress response to heat elicited by the heightened viscosity and thermal resistance of the medium [3]. Therefore, augmenting the concentration of RSM may foster the viability of LAB during spray drying.

In research publications, the primary factor leading to reduce the viability of LAB cells is cell membrane injury [7]. Consequently, the degree of cell membrane damage subsequent to spray drying is commonly explored to evaluate the potential of protective agents in enhancing viability [8,9,10]. Moreover, various methodologies have been employed to assess cell membrane properties, including detecting the damage ratio of membrane and examining the key enzyme activity variation in intramembrane [10,11]. A study by Cheng et al. (2022) demonstrated that the key enzymes associated with carbohydrate metabolism were impaired following the dehydration–drying process and that the injury to the membrane was reflected by the measurement of LDH and β-GAL activity [12]. Notably, HK, PK, LDH, and β-GAL play pivotal roles in carbohydrate metabolism pathways, particularly within the glycolytic pathways that provide energy for the improved growth of LAB strains [5,13]. From the perspective of energy generation and transformation view, β-GAL as glycoside hydrolase promotes the decomposition of polysaccharides into glucose or other monosaccharides [14]. Furthermore, HK and PK, as crucial enzymes responsible for carbohydrate utilization, contribute to the conversion of glucose into pyruvate, an important intermediate in the final stages of the LAB fermentation process. The conversion of pyruvate to lactic acid also represents a significant pathway for LAB to generate NAD and ATP under the catalysis of LDH [15]. Moreover, a study by Zhang et al. (2019) reported the utilization of β-GAL liposome model as a substitute for *Lactobacillus* cells in the optimization of a protective agent for spray drying [16]. Thus, the detriment to the cell membrane of LAB strains subjected to spray drying can be inferred through the assessment of alterations in these crucial enzymes involved in carbohydrate utilization, elucidating the manner in which effective protective agents enable LAB cells to withstand the rigors of spray drying and enhance their viability.

The glycolysis process and cell membrane transportation of LAB strains are not only for energy metabolism and nutrient material metabolism but also for the resistance to adverse external environments [17]. For instance, it has been reported that *Lactiplantibacillus plantarum* LIP-1 can withstand the stress of a long-term acid environment by up-regulating LDH enzyme genes associated with pyruvate metabolism and membrane-fatty-acid metabolic pathways [18]. Furthermore, in a study by Zhou et al. (2018), the addition of 5% *Lycium barbarum* polysaccharides as protective agents not only enhanced the viability of *Bifidobacterium* Bb-02 but also improved the activity of β-GAL, LDH, HK, and PK in bacterial cells exposed to freeze drying by overexpressing the genes of the transmembrane transport system (*opp*) and glycolysis process [19]. Evidently, the analysis of cells energy metabolism process of cells using transcriptomics technique should be instrumental in exploring the tolerance of LAB strains to spray drying.

Therefore, the objective of this study was designed to look for appropriate thermo-protective agents for selected LAB cells exposed to spray drying, with a focus on measuring the activity of carbohydrate utilization-related enzymes. Firstly, the study examined the detrimental effects of spray drying on the cell membranes of the LAB strains and their enzyme activity. Subsequently, it delved into understanding how spray drying produces injury in LAB cells through transcriptome data analysis. Lastly, the study utilized the enzymes LDH and PK as pivotal indicators to screen the desirable protective agents that improved the resistance of selected LAB strains to spray drying.

## 2. Materials and Methods

### 2.1. Materials

Recover skim milk (RSM) was purchased from Fonterra group Co., Ltd., Shanghai, China. One-step bacterial active protein extraction kit, Bicinchoninic Acid (BCA) protein assay kit, LDH assay kit, PK assay kit, β-GAL assay kit, and HK assay kit were bought from Suzhou Comin Biotechnology Co., Ltd., Suzhou, China. Live/Dead bacLight bacterial viability kit L7012, sodium glutamate, trehalose, and sucrose had all been prepared by Life technology Co., Ltd. (Tianjin, China) All other chemicals were analytical grade reagents and were prepared by Solarbio Co., Ltd. (Beijing, China).

### 2.2. Cultivation and Preparation of LAB Strains

*Lactobacillus acidophilus* NCFM (*Lb. acidophilus* NCFM), one of the common probiotic strains used to ferment dairy products, was purchased from DuPont Co., Ltd. (Shanghai, China). *Lactobacillus deribreckkii* subsp. *bulgaricus* CICC 6097 (*Lb*. *bulgaricus* CICC 6097) and *Lactiplantibacillus plantarum* CICC 21839 (*Lpb*. *plantarum* CICC 21839) were provided by the China Center of Industrial Microbial Culture Collection (CICC, Beijing, China).

Lyophilized powders of the 3 LAB strains were stored at −4 °C. Firstly, each strain was reactivated, and we restored their activity in 5 mL 12% (*w*/*v*) RSM, and each strain was anaerobically incubated at 37 °C for 18 h. Next, the strains were sub-cultured twice in 5 mL de Man–Rogosa–Sharpe medium (MRS) broth, which was incubated for 18 h at 37 °C to further activate. Then, the activated strains were inoculated in 100 mL MRS broth at 5% (*w*/*v*) inoculum volume and anaerobically incubated at 37 °C for 18 h. Subsequently, cells were collected through centrifugation (rotor radius was 15 cm, GL-20G-II, Anting Scientific Instruments Co., Ltd., Shanghai, China) at 5000 rpm/min for 4 °C at 10 min and the collected cells were washed twice with 0.01M PBS (pH 7.0). Then, the cells were resuspended in 0.5 mL 0.01M PBS (pH 7.0) to get cell suspensions, which were added to different protectant agents. Initial density of each bacterium was adjusted to10^10^ CFU/mL prior to use, and the initial density of each bacterium count was obtained via plate count.

### 2.3. Preparation of RSM Powders Containing LAB Strains

The 1% (*w*/*v*) or 10% (*w*/*v*) RSM medium was sterilized via autoclaving at 108 °C for 20 min prior to use. Next, the suspension of each LAB strain (0.5 mL), prepared as in Section 2.2, was added to 100 mL 1% (*w*/*v*) RSM and 100 mL 10% (*w*/*v*) RSM. The RSM solutions containing LAB cells were processed into spray-dried powders. Three groups were set up, namely 1% (*w*/*v*) RSM spray-dried powders containing live cells as 1% (*w*/*v*) RSM group, 10% (*w*/*v*) RSM spray-dried powders containing live cells as 10% RSM group, and 100 mL MRS medium containing LAB strains without experience spray drying as control group.

### 2.4. Conditions Setting for Spray Drying

The selected LAB strain powders were prepared using spray dryer (BUCHI Mini spray dryer B-290, BUCHI Co., Ltd., Geneva, Switzerland). Spray drying parameters were slightly modified according to Zhang et al. (2019) [16]. The inlet and outlet temperatures were 135 °C and 90 °C, respectively, and the feed flow rate was 7.5 mL/min at compressor air pressure 0.4 MPa, with an atomizer of a 0.6 mm nozzle to atomize protectant solution containing LAB strains. Finally, the powders were sealed in the convex bottle at 4 °C for further analysis.

### 2.5. Effects of RSM Protectants on the Viable Counts and Fermentation Activity of Selected LAB Strains

#### 2.5.1. Effects on the Viable Counts of Selected LAB Strains

The 0.5 g spray-dried powders of 3 selected LAB strains with protectants were dissolved in 5 mL sterile water to release cells. The cells were added into 10 mL sterile water to make a series of dilutions. Then, the 1.0 mL diluted suspensions were plated into MRS agar medium. The plates were incubated at 37 °C for 48 h to count viable cells. The viable counts of cells were presented as log CFU per gram of dried powders (log CFU/g) [4].

#### 2.5.2. Effects on the Fermentation Activity of Selected LAB Strains

Spray-dried powders of the selected 3 LAB strains with protectants (0.5 g) were dissolved into sterile water, and the suspensions were centrifugated at 5000 rpm/min for 10 min at 4 °C to collect the cells. The cells were inoculated into 100 mL RSM (12%, *w*/*v*) and incubated at 37 °C for 14 h. Next, pH values of the cultured milk, used to indicate the fermentation activity of each strain, were monitored using a pH meter (PHS-3C, Leici Co., Ltd., Shanghai, China) at a given interval with every 2 h (2, 4, 6, 8, 10, 12, and 14 h). The milk fermented by strains without having experienced spray drying was used as the control. The control group’s strains were inoculated into 100 mL RSM (12%, *w*/*v*) incubated at 37 °C for 14 h to detect their pH value every 2 h.

### 2.6. Enzymatic Activity Analysis

#### 2.6.1. Samples Preparation and Determination of Protein Content

The 0.5 g spray-dried powder was re-hydrated in 5 mL sterile water and then centrifugated at 5000 rpm/min for 10 min at 4 °C to harvested bacterial cells. Firstly, the protein of bacterial cells was extracted according to the instructions’ steps of kit. Then, the extracted proteins were quantified with BCA protein assay kit according to the instructions’ steps. The control group comprised LAB strains without having experienced spray drying. Secondly, the enzymatic activity was determined using relevant enzyme kits according to manufacturer’s instructions. One unit of enzyme activity (U) was expressed as 1 nmol per mg protein per minute (nmol/min/mg prot) [19].

#### 2.6.2. Determination of HK Activity

HK activity of each encapsulated strain was determined with HK assay kit based on the report of Zhou et al. (2018) [19]. The LAB strains without having experienced spray drying were used as the control group, and each strain was inoculated into 5 mL MRS broth at 5% (*w*/*v*) inoculum size, then anaerobically incubated at 37 °C for 18 h. After that, cells were collected through centrifugation (5000 rpm/min, 4 °C, and 10 min), and the collected cells were re-hydrated in 5 mL sterile water for further analysis. Firstly, spray-dried powder (0.5 g) was re-hydrated in 5 mL sterile water, and the cells from 5 mL suspensions were treated using ultrasonic cell breaker (TL-650Y, Tenlin Co., Ltd., Yancheng, China) at 0 °C with 200 W. The conditions used for the cell suspensions were sonicated for 3 s, then paused for 10 s. The ultrasound treatment for each sample was repeated 30 times. Then, the ultrasonic treated suspensions were centrifugated with 8500 rpm/min at 4 °C for 10 min to obtain supernatant. The regents were added into supernatant according to kit’s operation steps for preparing two centrifuge tubes, a blank tube, and a measuring tube. The absorbance of each tube was determined using ultraviolet spectrophotometer at 340 nm. The HK activity was computed according to Equation (1). One unit of HK activity was described as generating 1 nmol of NADPH per minute after each mg of samples’ protein interaction with the matrix.
(1)$$HK\;(nmol/min/mg\;prot)=\frac{643\times \Delta A}{\mathrm{Samples}’\;\mathrm{protein}\;\mathrm{quantity}}$$

Here, 643 is conversion coefficient, $\Delta A={\Delta A}_{320s}-{\Delta A}_{20s}$, ${\Delta A}_{320s}$ is the absorbance of measuring tube at 320 s, ${\Delta A}_{20s}$ is the absorbance of measuring tube at 20 s, and the unit of samples’ protein quantity is mg/mL.

#### 2.6.3. Determination of β-GAL Activity

β-GAL activity of each encapsulated strain was determined with β-GAL assay kit [12]. The control group was prepared as described in Section 2.6.2. Firstly, spray-dried powder (0.5 g) was re-hydrated in 5 mL sterile water, and the cells from 5 mL suspensions were treated with ultrasonic cell breaker (TL-650Y, Tenlin Co., Ltd., Yancheng, China) at 0 °C with 200 W. Each cell suspension was sonicated for 3 s and then paused for 10 s. The ultrasound treatment for each sample was repeated 30 times. Then, the ultrasonic treated suspensions were centrifugated at 16,300 rpm/min for 5 min at 4 °C to obtain supernatant. Secondly, the regents were added into the supernatant according to kit’s operation steps for preparing four centrifuge tubes, i.e., control tube, blank tube, standard tube, and measuring tube. The absorbance of each tube was determined using ultraviolet spectrophotometer at 400 nm. The β-GAL activity was computed according to Equation (2). One unit of β-GAL activity was described as generating 1 nmol of p-nitrophenol per minute after each mg of samples’ protein interaction with the matrix.
(2)$$\beta \text{-}GAL\;(nmol/min/mg\;prot)=\frac{59.83\times (\Delta A+0.0027)}{\mathrm{Samples}’\;\mathrm{protein}\;\mathrm{quantity}}$$

Here, 59.83 and 0.0027 are conversion coefficients, $\Delta A={\;\Delta A}_{\mathrm{measuring}\;\mathrm{tube}}-{\Delta A}_{\mathrm{control}\;\mathrm{tube}}$, ΔA is difference in absorbance between the absorbance of measuring tube and the absorbance of control tube, and the unit of samples’ protein quantity is mg/mL.

#### 2.6.4. Determination of PK Activity

PK activity of each encapsulated strain was determined with PK assay kit as described by Li et al. (2016) [20]. The control group was prepared as described in Section 2.6.2. Firstly, spray-dried powder (0.5 g) was re-hydrated in 5 mL sterile water, and the cells from 5 mL suspensions were treated with ultrasonic cell breaker (TL-650Y, Tenlin Co., Ltd., Yancheng, China) at 0 °C with 200 W. Each cell suspension was sonicated for 3 s and then paused for 10 s. The ultrasound treatment for each sample was repeated 30 times. Then, the ultrasonic treated suspensions were centrifugated at 8500 rpm/min for 10 min at 4 °C to obtain supernatant. The regents were added into supernatant according to the kit’s operation steps for preparing two centrifuge tubes, a blank tube, and a measuring tube. The absorbance of each tube was determined using ultraviolet spectrophotometer at 340 nm. The PK activity was computed according to Equation (3). One unit of PK activity was depicted as consuming 1 nmol of NADH per minute after each mg of samples’ protein interaction with the matrix.
(3)$$PK\;(nmol/min/mg\;prot)=\frac{1608\times \Delta A}{\mathrm{Samples}’\;\mathrm{protein}\;\mathrm{quantity}}$$

Here, 1608 is conversion coefficient, $\Delta A={\Delta A}_{140s}-{\Delta A}_{20s}$, ${\Delta A}_{140s}$ is the absorbance of measuring tube at 140 s, ${\Delta A}_{20s}$ is the absorbance of measuring tube at 20 s, and the unit of samples’ protein quantity is mg/mL.

#### 2.6.5. Determination of LDH Activity

LDH activity of each encapsulated strain was determined with LDH assay kit [19]. The control group was prepared as described in Section 2.6.2. Firstly, spray-dried powder (0.5 g) was re-hydrated in 5 mL sterile water, and the cells from 5 mL suspensions were treated with ultrasonic cell breaker (TL-650Y, Tenlin Co., Ltd., Yancheng, China) at 0 °C with 200 W. Each cell suspension was sonicated for 3 s and then paused for 10 s. The ultrasound treatment for each sample was repeated 30 times. Then, the ultrasonic treated suspensions were centrifugated at 8500 rpm/min for 5 min at 4 °C to obtain supernatant. Secondly, the regents were added into the supernatant according to kit’s operation steps for preparing four centrifuge tubes, i.e., control tube, blank tube, standard tube, and measuring tube. The absorbance of each tube was determined using ultraviolet spectrophotometer (UV-7, Mettler Toledo Co., Ltd., Zurich, Switzerland) at 450 nm. The LDH activity was computed according to Equation (4). One unit of LDH activity was defined as generating 1 nmol of pyruvic acid per minute after each mg of samples’ protein interaction with the matrix.
(4)$$LDH\;(nmol/min/mg\;prot)=\frac{184\times \Delta A}{\mathrm{Samples}’\;\mathrm{protein}\;\mathrm{quantity}}$$

Here, 184 is conversion coefficient, $\Delta A={\Delta A}_{\mathrm{measuring}\;\mathrm{tube}}-{\Delta A}_{\mathrm{control}\;\mathrm{tube}}$, ΔA is difference in absorbance between the absorbance of measuring tube and the absorbance of control tube, and the unit of samples’ protein quantity is mg/mL.

### 2.7. Effects of Spray Drying on Cell Membrane Property

#### 2.7.1. Detection of Cell Membrane Fluidity

The cell membrane fluidity was characterized using fluorescence spectrophotometer (Hitachi F-7000, HITACHI Co., Ltd., Tokyo, Japan) according to previously published method [5]. Firstly, spray-dried powder of each strain (0.5 g) was dissolved in 5 mL 0.01 M PBS (pH 7.0) and then centrifugated at 5000 rpm/min at 4 °C for 10 min to collect cells. The 1,6-Diphenyl-1,3,5-hexatriene (DPH, 2 mL, 2 μmol/L) was added into the collected cells, which were left at 37 °C for 30 min in a dark condition. Then, the DPH solution containing cells was centrifugated at 8000 rpm/min at 4 °C for 6 min to remove the supernatant. Secondly, the harvested cells were mixed with 2 mL 0.01 M PBS (pH 7.0) for further analysis. The excitation and emission wavelengths were set up to 340 nm and 427 nm, respectively. The cell membrane fluidity was described using fluorescence polarization (*p*) and micro-viscosity (*η*), which were calculated by Equation (5) and Equation (6), respectively.
(5)$$p=\frac{I_{\mathrm{VV}}-I_{\mathrm{VH}}G}{I_{\mathrm{VV}}+I_{\mathrm{VH}}G};\;G=\frac{I_{\mathrm{VH}}}{I_{\mathrm{HH}}}$$
(6)$$\eta =\frac{2P}{0.46-2P}$$

Here, I_VV_ and I_VH_ are the components of light intensity emitted. I_VV_ stands for the light parallel to the direction of the polarized excitation light; I_VH_ symbolizes the light perpendicular to the direction of the polarized excitation light; I_HH_ represents the light perpendicular to the direction of the polarized emission light. G is correction factor.

#### 2.7.2. Detection of Cell Membrane Integrity

The harvested bacterial cells were stained with Live/Dead bacLight bacterial viability kit L7012. The kit was composed of two separated nucleic acid dyes, PI (propidium iodide) and SYTO-9. Dye PI only gets through the destroyed cell membrane or dead cells’ membrane and fluoresces red. Dye SYTO-9 can stain intact cell membrane and fluoresces green [18]. The PI and SYTO-9 stock solutions were diluted into 5 μmol/L and 30 μmol/L, respectively, and diluted dyes 0.1 mL PI and 0.1 mL SYTO-9 were mixed for standby. The 0.5 g spray-dried powders were dissolved in 5 mL 0.01M PBS (pH 7.0) and centrifugated (5000 rpm/min for 10 min at 4 °C). After that, the cells were re-dispersed with 1 mL 0.01M PBS (pH 7.0), then 3 μL dye mixture was mixed with 1 mL cell suspension and incubated at 37 °C for 15 min in a dark condition. The stained bacterial suspension (5 μL) was placed on slide to obtain the micro-images using fluorescence microscope (DM500, Leica Co., Ltd., Wetzlar, Germany) [21]. Three micro-images of fluorescence chromogenic area (5 cm × 5 cm) from the control and treatment groups for each strain were randomly sampled and statistically analyzed using the software ImageJ 1.53a. The cell membrane integrity was expressed as the percentage of the chromogenic area of red/green cells to the total co-chromogenic area of cells [21].

### 2.8. Morphological Observation of Spray-Dried Powder Containing LAB Strains

The morphology of spray-dried powder containing the selected 3 LAB strains was detected via Hitachi S-4800 SEM (HITACHI Co., Ltd., Tokyo, Japan). The powders were adhered to double-sided conductive adhesive. The images were observed at an accelerating voltage of 1.0 kV. About 100 particles from SEM images of each sample group were randomly selected for analyzing their diameters by using ImageJ 1.53a software. The distribution of particle size was depicted using Origin 9.1 software [22].

### 2.9. Effect of Spray Drying on the Gene Expression of Lb. acidophilus NCFM

The transcriptome sequencing of strain NCFM was determined by referring to Zhang et al.’s (2020) method with some modifications [23]. Firstly, the activated strain NCFM cells were inoculated into 1000 mL MRS broth at 5% (*w*/*v*) inoculum volume and were anaerobically incubated at 37 °C for 18 h. The cells were collected through centrifugation with 5000 rpm/min at 4 °C for 10 min. Ten grams of wet bacterial pellets was segmented into two parts. One part (5 g) of strain NCFM, after being mixed with 1% or 10% RSM, was spray-dried and immediately dissolved into 50 mL 0.01 M PBS (pH 7.0) to collect cell pellets through centrifugation at 5000 rpm/min at 4 °C for 10 min. Another part (5 g) of strain NCFM cells without spray drying was used as the control group. And the resulting pellets from the control group and spray-dried groups were immediately stored in liquid nitrogen for RNA extraction. After that, the cells’ total RNA was extracted using Trizol (Aladdin Biotechnology Co., Ltd., Shanghai, China). And then, the purity of extracted RNA was tested using Nano Drop 2000 (Thermo Fisher Scientific Co., Ltd., Waltham, MA, USA). The quality of RNA was detected using Agilent 2100 Bioanalyzer system (Agilent Technologies Co., Ltd., Santa Clara, CA, USA), and high-purity RNA with OD_260nm/280nm_ ≥ 1.8 was in conformity with sequencing criteria. Then, sequencing libraries with high quality RNA (around 300 bp) were constructed using NEBNext UltraTM RNA Library Prep Kit for Illumina (Illumina Co., Ltd., San Diego, CA, USA) according to the manufacturer’s instructions. Finally, the library preparations (150 bp at cDNA each end) were sequenced through Illumina hiseq2000 (Illumina Co., Ltd., San Diego, CA, USA) to collect ‘reads’. Using HISAT and Bowtie 2–2.2.3 system to compare ‘reads’ with referred genes, the gene expression level was calculated using RSEM system according to comparison results to verify two samples could be compared with each other [24]. Then, differentially expressed genes (DEGs) of three groups were analyzed with PossionDis method [25]. DEGs between different groups were calculated using GOseq R package to control the false discovery rate (FDR); FDR was corrected with *p* < 0.05 as the screening threshold. Then, DEGs related to GO functions and KEGG pathway were analyzed via GO terms (Gene ontology, http://www.geneontology.org/ accessed on 20 January 2024) and the KEGG pathway library (http://www.genome.jp/kegg/ accessed on 20 January 2024) [18], respectively.

### 2.10. Physiological Performances of Lb. acidophilus NCFM Cells Spray-Dried with Different Composite Protectants

#### 2.10.1. Preparation of Spray-Dried Powder Containing Strain NCFM Encapsulated with 4 Composite Protectants

Four composite protectants, after dissolved into distilled water, formed a final volume of 100 mL solution, which was autoclaved at 108 °C for 20 min prior to use. Then strain NCFM cells inoculated in 100 mL MRS broth at 5% inoculum volume and anaerobically incubated at 37 °C for 18 h were harvested through centrifugation (5000 rpm/min, 4 °C, and 10 min), and resuspended in 5 mL 0.01M PBS (pH 7.0). The 5 mL suspension of strain NCFM was mixed with the four composite protectants before spray drying. The prepared samples of four groups containing strain NCFM cells for spray drying were composed of 10% (*w*/*v*) RSM with 3% (*w*/*v*) sodium glutamate as composite protectant A (CP-A group), 10% (*w*/*v*) RSM with 3% (*w*/*v*) sucrose as composite protectant B (CP-B group), 10% (*w*/*v*) RSM with 3% (*w*/*v*) trehalose as composite protectant C (CP-C group), and 10% (*w*/*v*) RSM with 3% (*w*/*v*) sodium glutamate plus 3% (*w*/*v*) trehalose plus 3% (*w*/*v*) sucrose as composite protectant D (CP-D group). The 10% (*w*/*v*) RSM spray-dried powders containing LAB strains were used as the control group.

#### 2.10.2. Effect of Composite Protectants on PK and LDH Activity of Strain NCFM

PK and LDH activities of strain NCFM in the presence of four composite protectants were measured as per assay kit’s operation steps.

#### 2.10.3. Effect of Four Composite Protectants on the Viable Counts of Strain NCFM

Approximately 0.5 g of spray-dried powders of strain NCFM with different composite protectants, after dissolved into 5 mL sterile water, were centrifugated at 5000 rpm/min at 4 °C for 10 min to harvest cells. The cells were added into 10 mL sterile water to make a series of dilutions. Then, 1.0 mL of strain NCFM cell dilutions were plated into MRS agar medium and incubated at 37 °C for 48 h to count viable cells. The viable counts of cells were presented as log CFU per gram of dried powders (log CFU/g) [4].

#### 2.10.4. Effect of Composite Protectants on the Fermentation Activity of Strain NCFM

Approximately 0.5 g of spray-dried powders of strain NCFM with four composite protectants, after dissolved into 5 mL sterile water, were centrifugated with 5000 rpm/min at 4 °C for 10 min to collect cells. pH values of the milk fermented by strain NCFM were determined as described in Section 2.5.2.

### 2.11. Statistical Analysis

All experiments were at least repeated three times. The results were processed using Origin 9.1 and GraphPad Prism 9.0. Each experiment was expressed in the form of mean ± standard deviation (SD). Significance differences among treatments at *p* < 0.05 among results were compared using Duncan’s new multiple-range test of one-way (ANOVA) using SPSS 22.0 statistical software.

## 3. Results

### 3.1. The Live Counts and Fermentation Activity of the Spray-Dried Selected LAB Strains in the Presence of RSM

#### 3.1.1. Live Counts of Selected LAB Strains Subjected to Spray Drying

To evaluate how spray drying influences the viable counts of the selected LAB strains, i.e., *Lpb*. *plantarum* CICC 21839, *Lb. acidophilus* NCFM, and *Lb*. *bulgaricus* CICC 6097, in the presence of different RSM concentrations (1% and 10% *w*/*v*), the live cell counts are presented in Figure 1. The live counts of three tested LAB strains after spray drying decreased, depending on the strains. When the RSM mass concentration was increased from 1% (*w*/*v*) to 10% (*w*/*v*), the viable counts of strain NCFM were promoted from 6.80 lg CFU/g to 7.27 lg CFU/g. Increasing the RSM concentration from 1% (*w*/*v*) to 10% (*w*/*v*) could increase the live cell numbers of strains CICC 6097 and CICC 21839 from 7.38 to 7.82 lg CFU/g and from 7.85 to 8.44 lg CFU/g, respectively. Clearly, using 10% (*w*/*v*) RSM as a carrier agent could improve the live counts of three LAB strains against spray drying, as reported by a similar study by Li et al. (2011) [5].

#### 3.1.2. Fermentation Activity of the Selected LAB Strains Subjected to Spray Drying

pH changes in milk fermented by the tested LAB strains were used to indicate the fermentation activity of these microorganisms before and after spray drying because generally, the pH value represents the H^+^ concentration in the fermentation medium [26]. As shown in Figure 2, significant differences in the pH values of milk fermented with the different strains were observed between the control group and RSM (1% and 10% *w*/*v*) groups after 14 h incubation at 37 °C. The three selected LAB strains had better fermentation activity in the presence of 10% (*w*/*v*) RSM than 1% (*w*/*v*) RSM, and the pH value of milk fermented by strain NCFM was 4.22 ± 0.014; it was 4.66 ± 0.007 for milk fermented by strain CICC 6097 and 3.64 ± 0.021 for milk fermented by strain CICC 21839. Thus, using 10% (*w*/*v*) RSM as a protective agent not only alleviated the damage of these tested LAB strains caused by spray drying but also improved their fermentation performance compared to 1% (*w*/*v*) RSM.

### 3.2. The Micrographics and Membrane Properties of Selected LAB Strains

#### 3.2.1. Microparticle Morphology of the Spray-Dried Powders Containing LAB Strains

Scanning electron micrographs (SEMs) were used to observe the morphological changes in and distribution of particle sizes of spray-dried powders containing three selected LAB strains (Figure 3). It was shown that an increase in RSM mass concentration from 1% (*w*/*v*) to 10% (*w*/*v*) led the average particle size of spray-dried powders containing LAB strains to increase. The average particle size of spray-dried powders increased from 6.45 ± 2.14 μm to 12.25 ± 3.46 μm for strain NCFM, from 7.79 ± 2.53 μm to 14.35 ± 5.38 μm for strain CICC 6097, and from 6.54 ± 2.07 μm to 11.57 ± 5.13 μm for strain CICC 21839. Compared to the 1% (*w*/*v*) RSM group (Figure 3i,iii,v), a higher average particle size in the 10% (*w*/*v*) RSM group meant more cells that would be embedded, as confirmed in Figure 3ii,iv,vi. Similar results were found by Yin et al. (2022) [27]. Moreover, the shapes of the microparticles in 10% (*w*/*v*) RSM groups were spherical with some hollows and collapses, which were able to be attributed to the rapid evaporation of water during the spray drying process [27,28].

#### 3.2.2. Cell Membrane Integrity of Selected LAB Strains

To visualize the serious damaging of cells caused by heat and dehydration during spray drying, the cell membrane integrity of these selected strains was measured via fluorescence staining kit method. SYTO9 and PI dyes from the kits could label complete cell membranes and damaged cell membranes [18], respectively. As shown in Figure 4A, the selected LAB strains had a higher cell membrane integrity than their controls, which were subjected to spray drying, because almost cells showed green fluorescence (Figure 4A(a,d,g)). And the membrane integrity of the LAB strains was improved better when the concentration of RSM as a protectant increased from 1% to 10% (*w*/*v*) because an apparent decrease in the proportion of red fluorescence was observed (Figure 4A(b,c,e,f,h,i)). As shown in Figure 4B, the green cell chromogenic areas of selected LAB strains from control group were all above 97%, and the percentage of green cell chromogenic areas gradually increased as the concentration of RSM improved, but the red cell chromogenic areas gradually decreased. Regarding the ratio of green cell chromogenic areas, it gradually augmented from 36.20% ± 2.34 in the 1% (*w*/*v*) RSM group to 82.16% ± 0.57 in the 10% (*w*/*v*) RSM group due to strain NCFM, from 32.37% ± 2.41 to 85.29% ± 0.65 due to strain CICC 6097, and from 27.97% ± 2.05 to 64.86% ± 2.21 due to strain CICC 21839. Thus, the selection of 10% (*w*/*v*) RSM as a basic protectant should help the selected LAB strains tolerate spray drying better.

#### 3.2.3. Cell Membrane Fluidity of the LAB Strains

The cell membrane fluidity values of the selected three LAB strains treated with or without spray drying were described through fluorescence polarization (*p*) and micro-viscosity (*η*) values. It is interesting to note that the bigger the values of *p* and *η* are, the smaller the fluidity of the cell membrane is [5]. As shown in Table 1, there was a big difference in membrane fluidity values among the tested three LAB strains before and after spray drying (*p* < 0.05). The *P* and *η* values of membrane fluidity for the three selected LAB from 10% (*w*/*v*) RSM group were all higher than in the control group, being increased by 1.07-fold and 1.20-fold for strain NCFM, 1.06-fold and 1.19-fold for strain CICC 6097, and 1.04-fold and 1.14-fold for strain CICC 21839. It was indicated that the membrane fluidity of strains after spray drying was more susceptible to damage than in the control group.

In summary, the morphology and cell membrane properties of the three selected LAB strains were greatly affected by spray drying, but 10% (*w*/*v*) RSM as a carrier could improve cells’ tolerance to spray drying.

### 3.3. The Enzyme Activity Related to Carbohydrate Utilization of LAB Strains

To clarify how such an extreme environment as spray drying impacts the carbohydrate utilization-related enzymes, the activities of HK, β-GAL, PK, and LDH from the selected LAB strains were determined as follows.

#### 3.3.1. HK Activity

As seen in Figure 5a, a significant decline in HK activity in these LAB strains after experiencing spray drying was observed compared to their control group (without spray drying). Compared to that of the 10% (*w*/*v*) RSM group, the HK activity of the control group was higher 0.51-fold for strain NCFM, 0.92-fold for strain CICC 6097, and 0.72-fold for strain CICC 21839. However, although the HK activity of the three LAB strains in the 10% (*w*/*v*) RSM group was slightly higher than that of the 1% (*w*/*v*) RSM group, both strains, NCFM and CICC 6097, did not show a significant difference in their HK activity after experiencing spray drying. This meant that the selection of HK activity as a sensitive indicator of the tested LAB strains to spray drying was not quite desirable for the present work. Thus, the HK activity could not serve as a crucial enzyme indicating the efficiency of a protective agent that was used to improve the tolerance of the tested LAB strains to spray drying in following experiments.

#### 3.3.2. β-GAL Activity

As shown in Figure 5b, compared to the control, the β-GAL activity of the three tested LAB strains after spray drying significantly decreased for both the 1% (*w*/*v*) RSM and 10% (*w*/*v*) RSM groups (*p* < 0.05). However, strain NCFM after spray drying lost its β-GAL activity more than the other two LAB strains. The results showed that the strain NCFM membrane suffered from great damage and leakage due to heat stresses during spray drying as β-GAL is an intracellular enzyme [29]. The β-GAL activity of the three tested LAB strains in the 10% (*w*/*v*) RSM group was observed to be higher than in the 1% (*w*/*v*) RSM group after spray drying. However, a significant difference in β-GAL activity between strains CICC 6097 and CICC 21839 subjected to spray drying only took place when the RSM concentration increased from 1% to 10% (*w*/*v*) (*p* < 0.05). It was clear that β-GAL had not guaranteed the sensitivity of all the tested LAB strains to spray drying very well. Therefore, β-GAL could not be quite good as one of the carbohydrate utilization-related key enzymes for choosing a valid spray-drying protective agent in subsequent experiments.

#### 3.3.3. PK Activity

As shown in Figure 5c, significant differences in the PK activity of the selected three LAB strains, after being exposed to spray drying, were detected among all experimental groups (*p* < 0.05). And the PK activity of the tested three LAB strains in the 10% (*w*/*v*) RSM group was evidently higher than that in the 1% (*w*/*v*) RSM group (*p* < 0.05): about 1.29-fold for strain NCFM, 1.73-fold for strain CICC 6097, and 2.14-fold for strain CICC 21839. It was clear that the PK activity of the selected three LAB strains would be more susceptible to spray drying. Hence, PK should be ideal as one of the key carbohydrate utilization-related enzymes, indicating the efficiency of a protective agent that was used to improve the tolerance of the tested LAB strains to spray drying in following experiments.

#### 3.3.4. LDH Activity

As seen in Figure 5d, the LDH activity of the three selected LAB strains reduced after spray drying but was promoted by 3.04-fold for strain NCFM, 1.52-fold for strain CICC 6097, and 1.31-fold for stain CICC 21839 when the RSM concentration increased from 1% to 10% (*w*/*v*). Moreover, the LDH activity of the selected LAB strains, after being exposed to spray drying, showed significant differences among all experimental groups (*p* < 0.05). Clearly, the LDH activity of the selected three LAB strains would be more susceptible to spray drying. Thus, LDH also should work as one of the key carbohydrate utilization-related enzymes to indicate the protective efficiency of an agent carrying LAB strains subjected to spray drying in the following experiments.

Regarding four tested key carbohydrate utilization-related enzymes, our present results showed the enzymes PK and LDH from the selected LAB strains responded to spray drying more sensitively than HK and β-GAL. At the same time, strain NCFM showed a poor tolerance to spray drying compared to other LAB strains. Comprehensively, we picked up strain NCFM as the tested strain to further investigate the damaged mechanism of spray drying towards cells using the transcriptomics method, referring to the study of Corcoran et al. (2004) [30].

### 3.4. The Transcriptomics of Lb. acidophilus NCFM Subjected to Spray Drying in the Presence of RSM

#### 3.4.1. Identification of Differential Gene Expression (DEGs)

To explore the possible effect of heat and dehydration on injured cells, the related DEGs of strain NCFM subjected to spray drying were determined in the presence of RSM (1% and 10%, *w*/*v*) and were compared with its cells without having experienced spray drying. As shown in Figure 6A, regarding the control group vs. the 1% RSM group, the annotated 664 DEGs included 205 up-regulated genes and 459 down-regulated genes. Regarding the 1% (*w*/*v*) RSM group vs. the 10% (*w*/*v*) RSM group, a total of 153 genes were annotated, including 118 up-regulated genes and 35 down-regulated genes.

#### 3.4.2. Annotation of the GO Database

The pairwise comparison of each group is shown in Figure 6B(a), and the GO database was applied to sort these transcripts. The genes sorted from GO database analysis were mainly classified into three categories, i.e., biological processes (BPs), cellular components (CCs) and molecular functions (MFs). Regarding the 1% (*w*/*v*) RSM group vs. the control group, the BP category consisted of seventeen entries containing a total of 303 annotated genes; the CC function was composed of four entries with a total of 53 annotated genes; and the MF category had fourteen entries, containing a total of 308 annotated genes. Mostly, the annotated genes were enriched in BPs responsible for the phosphoenolpyruvate phosphotransferase system, the carbohydrate metabolic process, and membrane metabolism in terms of GO database analysis. The results from the GO database analysis implied the significant damage of spray drying towards the sugar metabolic process and membrane function of strain NCFM at a molecular level.

Regarding the 1% (*w*/*v*) RSM group vs. the 10% (*w*/*v*) RSM group, as shown in Figure 6B(b), the BP category was composed of fifteen entries, containing a total of seventy-six annotated genes; the CC function was made up of three entries, with a total of six annotated genes; and the MF category had seventeen entries, containing a total of seventy-one annotated genes. Mostly, the annotated genes were enriched in MFs responsible for the transmembrane transport process, oxidoreductase system, and DNA structure system in terms of GO database analysis. At a molecular level, increasing the numbers of up-regulated genes from the 10% (*w*/*v*) RSM group vs. the 1% (*w*/*v*) RSM group illustrated that the increase in RSM content should have a positive “buffer effect” on the metabolisms of strain NCFM damaged in spray drying.

#### 3.4.3. KEGG Pathways

To explore how the DEG expression works across different treatment groups, the protein enzymes of strain NCFM involved in the metabolic pathway were used for KEGG functional annotation (Table 2). Compared to the control group, the ABC transporters’ pathway was up-regulated in the 1% (*w*/*v*) RSM group. The ABC transporter family is a large class of transmembrane transport proteins that can transfer their binding substrate into or out of a membrane on the premise of providing transmembrane transport energy through ATP hydrolase [31]. The results showed that the tolerance of strain NCFM to spray drying was improved by accelerating the rate of material transport due to the presence of RSM. However, it was noted that most genes that are mainly enriched in sugar metabolism, the glycolysis process, and cell membrane functions were down-regulated in the control group vs. the 1% (*w*/*v*) RSM group. Firstly, the down-regulation of sucrose-6-phosphate hydrolase (*scrB*) and β-phosphoglucomutase (β-PGM) (*pgmB*), which are catabolic enzymes responsible for sucrose and trehalose metabolism pathways [32,33], showed that spray drying had reduced the utilization ability of LAB to sucrose and trehalose. Therefore, supplements of sucrose and trehalose as the components of composite protectants to RSM might produce positive protection for improving the viability of selected bacterial cells exposed to spray drying. Secondly, the down-regulation of the glycolysis pathway and cysteine metabolism pathway directly impaired the expressions of pyruvate synthase (*pyk*) encoding PK, as well as cystathionine β-lyase (*metC*), which generated more pyruvate to enter into the glycolysis process [18,31]. At the same time, the expression of L-lactate dehydrogenase (*ldh*) was down-regulated, causing the imbalance of the pyruvate metabolism pathway [15]. Obviously, the down-regulated expressions of pyruvate synthase (*pyk*) and L-lactate dehydrogenase (*ldh*) led the PK and LDH activity of strain NCFM after spray drying to be decreased, as observed in Section 3.3.3 and Section 3.3.4. Thirdly, it was seen that the expression of the glycerolipid metabolism pathway responsible for dihydroxyacetone kinase subunit L (*dhal, dhaK*) and glycerol kinase (*glpK*) was down-regulated, indicating that the participation membrane synthesis process and glycerol-3-phosphate (G3P) synthesis process were inhibited [19], as shown in Section 3.2.2. Therefore, the key enzymes related to carbohydrate metabolism and the cell membrane function of strain NCFM were seriously damaged by spray drying, posing a vital threat to its cells’ viability.

Regarding the 10% (*w*/*v*) RSM group vs. the 1% (*w*/*v*) RSM group, the up-regulated genes of strain NCFM were mainly expressed in the oxidative stress process, sugar metabolism, and membrane transport function. Firstly, the thioredoxin-disulfide reductase (*trxB*) as an important component of the thioredoxin system was up-regulated, indicating the inhibition of cell apoptosis throughout regulating the strains’ oxidative stress during spray drying. Generally, sodium glutamate is often chosen as one of the desirable ingredients to protect the membrane protein from oxidative stress during spray drying, and 3% (*w*/*v*) sodium glutamate has been found to significantly improve the viability of LAB strains in extreme environment [16,17]. Thus, the supplementation of 3% (*w*/*v*) sodium glutamate as a protective agent to RSM might improve the oxidative stress of bacterial cells to spray drying. Secondly, the expression of the ABC transporter pathway responsible for the glycerol-3-phosphate (G3P) ABC transporter (*ugpC*) was up-regulated, promoting G3P transported into the cell membrane synthesis process or glycolytic pathway and resultantly sustaining such membrane functions as integrity and fluidity [34]. Thirdly, the phosphotransferase system (PTS) pathway involved in lactose/cellobiose transporter subunit IIA (*celC*) was up-regulated, improving phosphorylated lactose and cellobiose, which are decomposed monosaccharides entering the glycolysis process and thus providing the energy for strain NCFM to resist the severe environment [31]. Obviously, the increase in RSM concentration from 1% (*w*/*v*) to 10% (*w*/*v*) should be beneficial for the improved tolerance of strain NCFM to spray drying.

Briefly, the expressions of genes encoding PK (*pyk*) and LDH (*ldh*), which are related to the glycolysis process, were significantly down-regulated in terms of strain NCFM transcriptomics after spray drying. An increase in RSM concentration from 1% (*w*/*v*) to 10% (*w*/*v*) not only significantly improved the ABC transporter and PTS system related to cell membrane function but also let strain NCFM cells obtain more resistance to adverse environments like spray drying. Especially, it was speculated that the addition of sucrose, trehalose, or sodium glutamate as one of the assisted protectants to RSM might benefit the performance of catabolic pathways and oxidative stress of the tested cells exposed to spray drying.

### 3.5. Effects of RSM-Based Composite Protective Agents on the Physiological Properties of Lb. acidophilus NCFM

#### 3.5.1. Effects of RSM-Based Composite Protective Agents on PK and LDH Activity of Strain NCFM

The transcriptomics results of strain NCFM showed us that the supplementation of other protective ingredients to RSM (10%, *w*/*v*) should be beneficial for the improved resistance of the bacterial cells to spray drying. Thus, four groups of RSM-based composite protective agents, including the sodium CP-A group (RSM plus glutamate), CP-B group (RSM plus sucrose), CP-C group (RSM plus trehalose), and CP-D group (RSM plus glutamate, sucrose, and trehalose), were designed to examine their potential in protecting *Lb. acidophilus* NCFM from the physiological damage of spray drying. Firstly, as shown in Figure 7a,b, using CP-D as a composite protective agent provided the best protection for the PK and LDH activity of strain NCFM after spray drying compared to the other test groups. The PK activity of strain NCFM was 1.43-fold higher than that of the CP-A group, 1.37-fold higher than that of the CP-B group, 1.13-fold higher than that of the CP-C group, and 1.42-fold higher than that of the RSM group. The LDH activity of strain NCFM was 1.43-fold higher than that of the CP-A group, 2.41-fold higher than that of the CP-B group, 1.47-fold higher than that of the CP-C group, and 3.33-fold higher than that of the RSM group.

Obviously, PK and LDH could be considered as carbohydrate utilization-related enzymes to indicate the efficiency of a protective agent in preserving cells against spray drying.

#### 3.5.2. Effect of RSM-Based Composite Protective Agents on the Viable Counts of Strain NCFM

As seen in Figure 7c, the CP-D group preserved strain NCFM subjected to spray drying better than other groups. The live cell counts of strain NCFM after spray drying reached 8.56 log CFU/g for the CP-D group, followed by 7.95 log CFU/g for the CP-A group, 7.86 log CFU/g for the CP-C group, 7.55 log CFU/g for the CP-B group, and 7.28 log CFU/g for the RSM group. Clearly, the use of CP-D as a composite protective agent had significantly improved the tolerance of strain NCFM to spray drying, producing the best viability.

#### 3.5.3. Effect of RSM-Based Composite Protective Agent on the Fermentation Performance of Strain NCFM

As shown in Figure 7d, compared to other groups, the CP-D group favored best in terms of the fermentation performance of strain NCFM, which was indicated by pH changes in fermented milks. After 14 h incubation, the lowest pH of 3.7 ± 0.01 of fermented milk in the CP-D group was observed compared to other groups. Obviously, the use of CP-D group as a composite protective agent not only protected strain NCFM better but quickly recovered its fermentation activity from the spray drying stress as well.

## 4. Discussion

Rapid changes in drop temperature and water content during spray drying will cause various injuries towards LAB strains, including damages from heat and oxidative stresses. Various types of stress destroy multiple parts of cells such as cell membranes and key enzyme activities [6,35]. Numerous studies have reported the great potential of RSM in protecting LAB cells against adverse environments [3,36]. RSM has been found to improve the heat resistance of *Lacticaseibacillus rhamnosus* GG to spray drying [35]. Research conducted by Cheng et al. (2022) reported that 10% (*w*/*v*) RSM effectively prevented cell membrane damage and intracellular substance leakage since RSM may form a protein film outside bacteria, increasing the LDH and β-GAL activity of *Lactiplantibacillus plantarum* L1 and *Limosilactobacillus fermentum* L2 [12]. In our study, 10% (*w*/*v*) RSM was more conducive to improving the viability of LAB strains compared to 1% (*w*/*v*) RSM when testing them as carrier agents of spray drying (Figure 1). The potential mechanism may be related to the protection of cell membrane and key enzyme activity by highly concentrated RSM.

The structure and function of cell membrane play an important role in the growth and nutrient metabolism of LAB strains. Bacteria have usually developed several adaptive mechanisms including membrane integrity and fluidity adjustment to cope with stress environments [37]. In our work, the membrane integrity and fluidity of the selected LAB strains subjected to spray drying were preserved better in the 10% (*w*/*v*) RSM group than in the 1% (*w*/*v*) RSM group (Figure 4 and Table 1), indicating that high contents of RSM decreased the cellular contents’ leak risk. At the same time, the activity of LDH and β-GAL, two carbohydrate utilization-related enzymes, could be well sustained in the presence of 10% (*w*/*v*) RSM (Figure 5b,d). Moreover, the live cell counts of three selected strains were all above 10^7^ CFU/g when the concentration of RSM increased from 1% (*w*/*v*) to 10% (*w*/*v*) (Figure 1). It is obvious that a high concentration of RSM should be beneficial for the improvement of cell membrane properties, enhancing the ability of the tested strains to resist stresses and prolong their living time during spray drying [38]. In addition, a high concentration of RSM allows the tested strains to respond to stress environments by adjusting their genetic expression related to cell membranes [3]. Our results from transcriptomic analysis indicated the up-regulation expression of transmembrane transporter family (ABC transporters) and the PTS family protein in the presence of 10% (*w*/*v*) RSM, presumably allowing strain NCFM to resist harsh environments by utilizing the energy generated by ATP hydrolysis (*MdlB*) and changing the fluidity of its cell membrane (Table 2) [39]. At the same time, the presence of 10% (*w*/*v*) RSM up-regulated the glycerol-3-phosphate ABC transporter (*ugpC*) of strain NCFM, further indicating the improvement of the cell membrane integrity due to promoting the synthesis of membrane fatty acids during spray drying. Therefore, using 10% (*w*/*v*) RSM as a carrier acted as an effective protective barrier for the bacterial strains, reducing the loss of membrane function caused by various stresses during spray drying.

The key enzymes stemming from the carbohydrate metabolism in LAB strains are mainly response for the glycolysis pathway and have been widely investigated [40]. In our case, higher activities of four enzymes, HK, β-GAL, PK, and LDH, in the selected LAB strains were maintained in the presence of 10% (*w*/*v*) RSM (Figure 5). Particularly, we used transcriptomic data of strain NCFM to describe the carbohydrate metabolism processes responsible for the glycolytic pathway since the live cell counts and cell membrane properties of strain NCFM undergoing spray drying were not as good as those of strains CICC 6097 and CICC 21839. Strain NCFM down-regulated such glycolysis pathways as the pyruvate synthase (*pyk*)-encoding PK protein and L-lactate dehydrogenase (*ldh*)-encoding LDH protein pathways in the presence of 1% (*w*/*v*) RSM during spray drying [41,42], but the presence of 10% (*w*/*v*) RSM significantly up-regulated the PTS transporter and pyruvate metabolism of strain NCFM. The up-regulated PTS transporter and pyruvate as an intermediate link of energy and sugar metabolism should be quite beneficial for the improvement of strain NCFM’s carbohydrate utilization, which produces enough energy to respond to a hostile environment due to the enhancement of enzyme activity [15,42]. The transcriptomic data from strain NCFM explain how the key glycolytic enzymes PK and LDH respond to spray drying and perhaps could be used as key indicators to screen the desirable protective agents for bacterial cells.

The oxidation–reduction process and expression genes responsible for the sucrose (*scrB*) and trehalose (*pgmB*) utilization of strain NCFM were significantly down-regulated after this bacterium experienced the stress of spray drying (Table 2 and Figure 6). Thus, we inferred that the supplementation of sodium glutamate, sucrose, and trehalose as ingredients in 10% (*w*/*v*) RSM to construct composite protective agents might help the selected bacterial strains challenge spray drying, improving the fermentation activity of the injured cells better. In our case, the potential of 10% (*w*/*v*) RSM in combination with sucrose, trehalose, and sodium glutamate as composite protective agents in improving the viability and key enzyme activity of strain NCFM was examined. It was found that the different combinations of 10% (*w*/*v*) RSM with sodium glutamate, sucrose, and trehalose provided different protections for strain NCFM cells subjected to spray drying. The CP-D group, consisting of 10% (*w*/*v*) RSM, 3% (*w*/*v*) sodium glutamate, 3% (*w*/*v*) sucrose, and 3% (*w*/*v*) trehalose as a composite agent, provided the best protection for strain NCFM, with viable counts of 8.56 log CFU/g compared to the other groups (Figure 7). Sucrose and trehalose have been reported to not only form hydrogen bonds with cell membrane proteins but also tend to remain glassy in a dry state, stabilizing cell components and providing strong thermal protection for strains during spray drying [36]. Sodium glutamate is often used as a common amino acid protectant to stabilize the cell membrane proteins [16]. Resultantly, the screened composite protectant composed of RSM plus sucrose, trehalose, and sodium glutamate, as indicated by the LDH and PK enzyme activities of strain NCFM, not only effectively preserved the viability of this microorganism but also enhanced its fermentation performance after spray drying.

## 5. Conclusions

The present work was designed to find a close connection between the four carbohydrate utilization-related enzymes of selected LAB strains and their RSM-based protective agents available for spray drying. An increase in RSM concentration from 1% (*w*/*v*) to 10% (*w*/*v*) as a basic carrier of several selected LAB strains not only significantly maintained cell membrane properties but also preserved the enzyme activity very well, greatly improving the viability and fermentation performance of these bacterial cells subjected to spray drying. Transcriptomic data obtained from strain NCFM cells, which were used in a case study, confirmed that the two enzymes PK and LDH were significantly affected by spray drying due to the down-regulated pyruvate synthase (*pyk*) and L-lactate dehydrogenase (*ldh*). As a result, according to the enzyme activities of LDH and PK from strain NCFM, a composite protectant composed of 10% (*w*/*v*) RSM plus 3% (*w*/*v*) sucrose, 3% (*w*/*v*) trehalose, and 3% (*w*/*v*) sodium glutamate was intentionally designed and confirmed to effectively protect this microorganism’s cells against spray drying. The live cell counts of strain NCFM experiencing spray drying even reached 10^8^ CFU/g, and its fermentation performance in milk was largely improved compared to 10% (*w*/*v*) RSM as a single protective agent or in combinations with sucrose, trehalose, or sodium glutamate. This is the first report showing that the carbohydrate utilization-related key enzymes PK and LDH could be used as sensitive indicators to screen desirable protective agents that are able to improve the tolerance of the selected LAB strains to harsh conditions like spray drying. It should be emphasized that more studies still need to be conducted to elucidate the feasibility of LDH and PK enzymes as sensitive indicators in selecting the wanted protectants for strains subjected to spray drying.

## Figures and Tables

**Figure 1 microorganisms-12-01094-f001:**
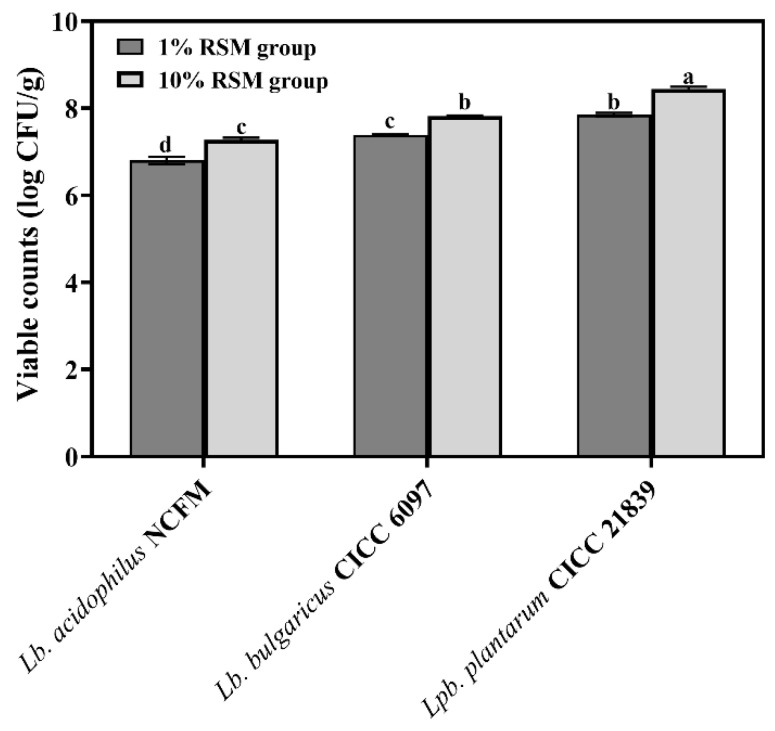
The viable counts (log CFU/g) of spray drying powders containing *Lb. acidophilus* NCFM, *Lb*. *bulgaricus* CICC 6097, and *Lpb*. *plantarum* CICC 21839 from 1% (*w*/*v*) RSM group or 10% (*w*/*v*) RSM group. Different letters from a to d on the top of each column indicate significant differences (*p* < 0.05).

**Figure 2 microorganisms-12-01094-f002:**
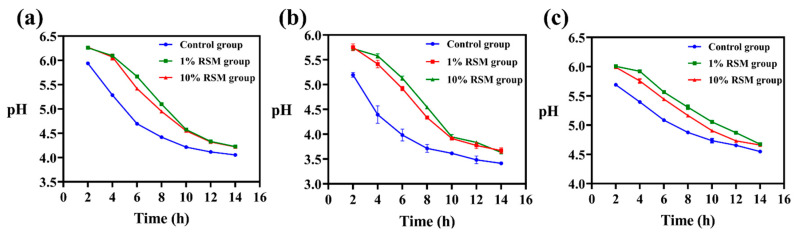
Images (**a**–**c**) show pH changes in milk fermented by *Lb. acidophilus* NCFM, *Lb*. *bulgaricus* CICC 6097, and *Lpb*. *plantarum* CICC 21839 after spray drying, respectively.

**Figure 3 microorganisms-12-01094-f003:**
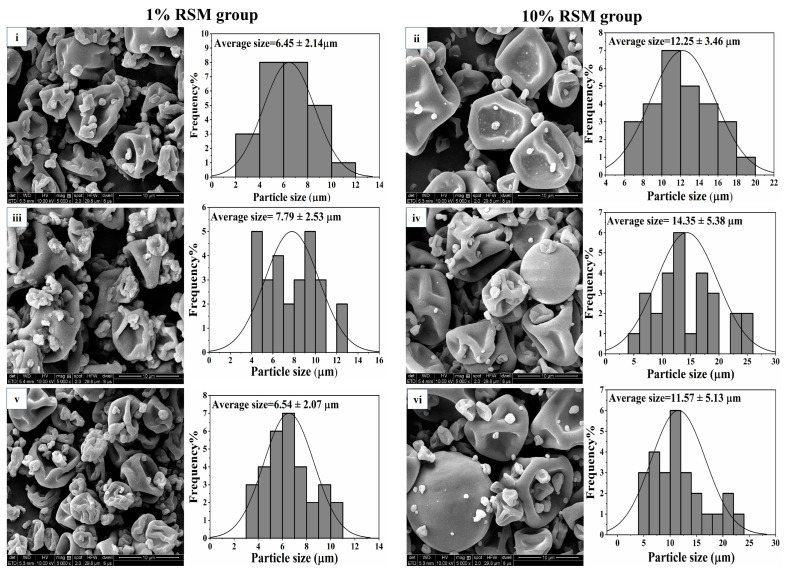
Microscopic characteristics of powders and membrane integrity of cells after spray drying. Image A shows the morphology and particle size distributions of spray-dried powders containing the selected 3 LAB strains. Images (**i**,**iii**,**v**) are the morphological changes in 1% (*w*/*v*) RSM group powders with *Lb. acidophilus* NCFM, *Lb*. *bulgaricus* CICC 6097, and *Lpb*. *plantarum* CICC 21839, respectively (magnification = 5000×). Images (**ii**,**iv**,**vi**) are the morphological changes in 10% (*w*/*v*) RSM group powders with *Lb. acidophilus* NCFM, *Lb*. *bulgaricus* CICC 6097, and *Lpb*. *plantarum* CICC 21839, respectively (magnification = 5000×). Particle size distribution images on the right of each SEM image represent the average particle size of each powder.

**Figure 4 microorganisms-12-01094-f004:**
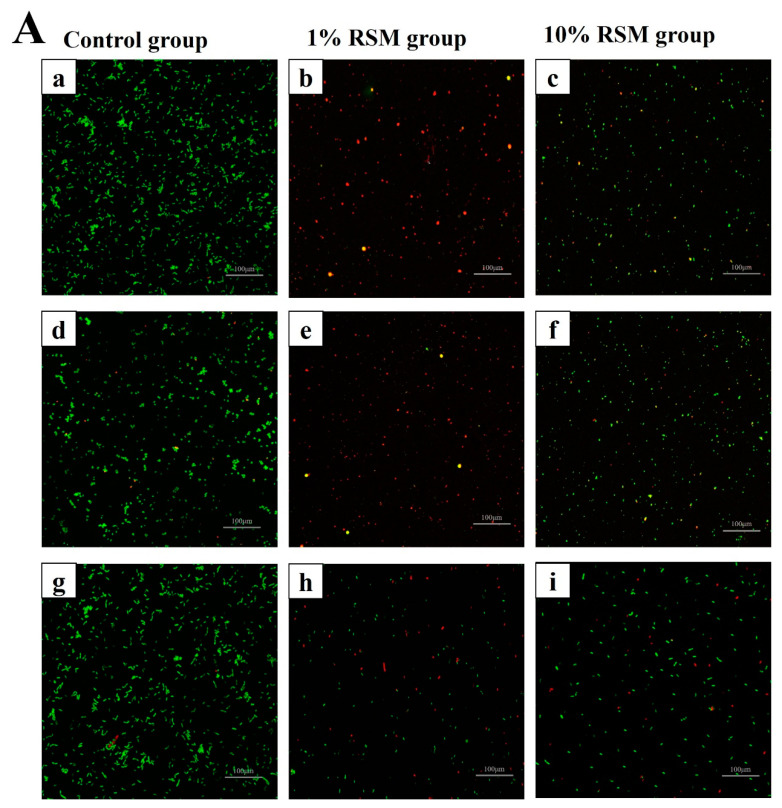
Image (**A**) shows the cell membrane integrity of the selected 3 LAB strains after spray drying. Images (**a**,**d**,**g**) from control group indicate 3 LAB strains without having experienced spray drying. Images (**a**–**c**) represent *Lb. acidophilus* NCFM from control group, 1% (*w*/*v*) RSM group, and 10% (*w*/*v*) RSM group, respectively. Images (**d**–**f**) represent *Lb*. *bulgaricus* CICC 6097 from control group, 1% (*w*/*v*) RSM group, and 10% (*w*/*v*) RSM group, respectively. Images (**g**–**i**) represent *Lpb*. *plantarum* CICC 21839 from control group, 1% (*w*/*v*) RSM group, and 10% (*w*/*v*) RSM group, respectively. Image (**B**) shows relative fluorescent intensities from different strains groups’ fluorescence microscope images. The red/green color represents the percentage of the chromogenic area of red/green cells to the total co-chromogenic area of cells in the fluorescence microscope images. Image (**a**–**c**) represent *Lb. acidophilus* NCFM, *Lb*. *bulgaricus* CICC 6097, and *Lpb*. *plantarum* CICC 21839, respectively.

**Figure 5 microorganisms-12-01094-f005:**
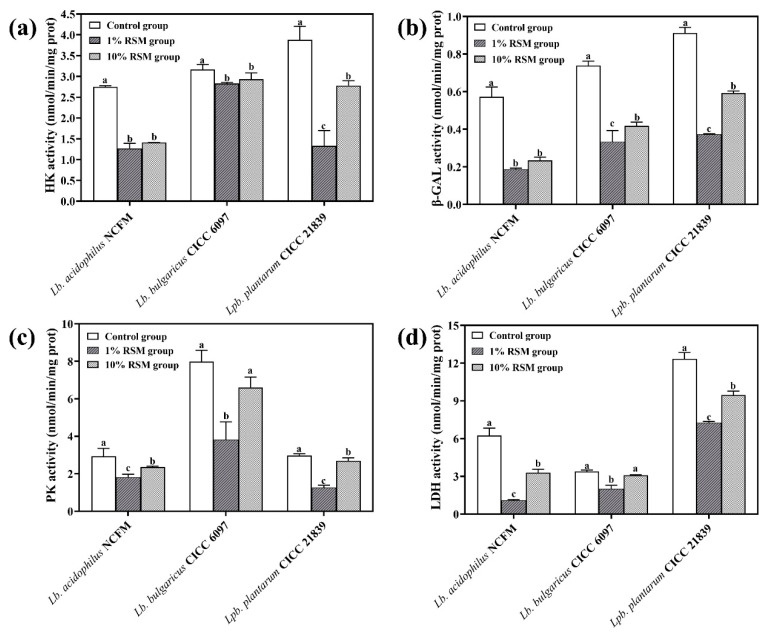
The four key enzyme activities of *Lb. acidophilus* NCFM, *Lb*. *bulgaricus* CICC 6097, and *Lpb*. *plantarum* CICC 21839 from 1% (*w*/*v*) RSM group or 10% (*w*/*v*) RSM group. The control group results indicate the 3 LAB strains without having experienced spray drying. Images (**a**–**d**) show HK activity, β-GAL activity, PK activity, and LDH activity of 3 LAB strains in 1% (*w*/*v*) RSM group or 10% (*w*/*v*) RSM group. Different letters from a to c on the top of each column indicate significant differences (*p* < 0.05).

**Figure 6 microorganisms-12-01094-f006:**
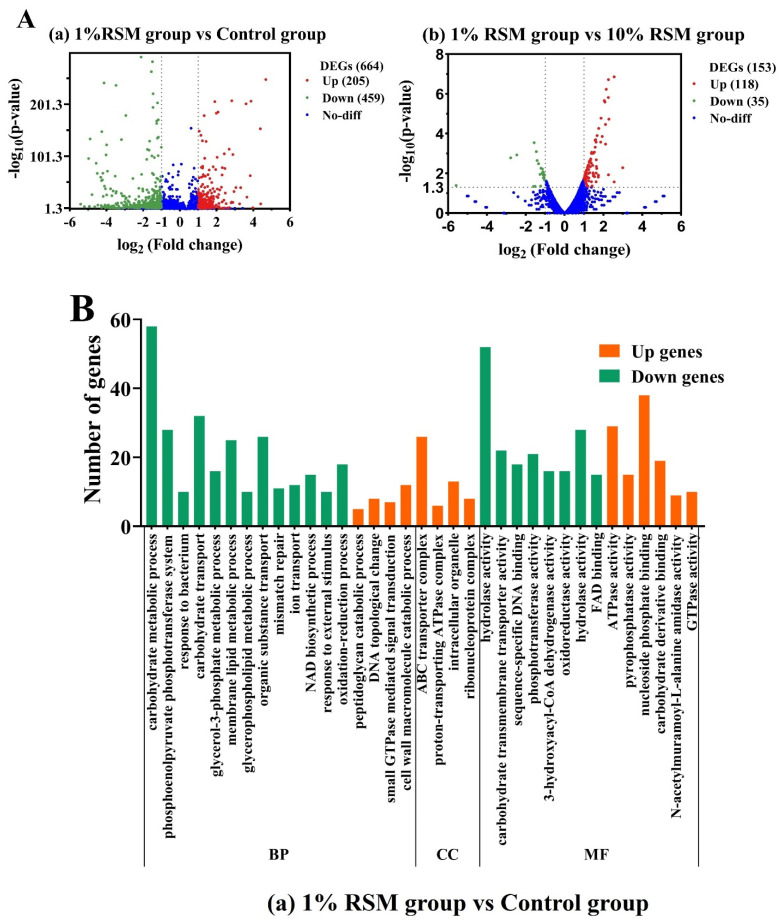
(**A**) Volcano plot of differential genes in *Lb. acidophilus* NCFM. The red pots represent up-regulated differential genes, green pots represent down-regulated differential genes, and blue pots represent no differential genes. (**B**) The number of different clusters of expressed genes in the GO terms enriched by the biological process (BP), molecular function (MF), and cell component (CC) in *Lb. acidophilus* NCFM. The differential genes of control group vs. the differential genes of 1% (*w*/*v*) RSM group mean the expressed genes of *Lb. acidophilus* NCFM from 1% (*w*/*v*) RSM group vs. the expressed genes of *Lb. acidophilus* NCFM without having undertaken spray drying (*p* < 0.05); the differential genes of 10% (*w*/*v*) RSM group vs. the differential genes 1% (*w*/*v*) RSM group indicate the expressed genes of *Lb. acidophilus* NCFM from 10% (*w*/*v*) RSM group vs. the expressed genes of *Lb. acidophilus* NCFM from 1% (*w*/*v*) RSM group (*p* < 0.05).

**Figure 7 microorganisms-12-01094-f007:**
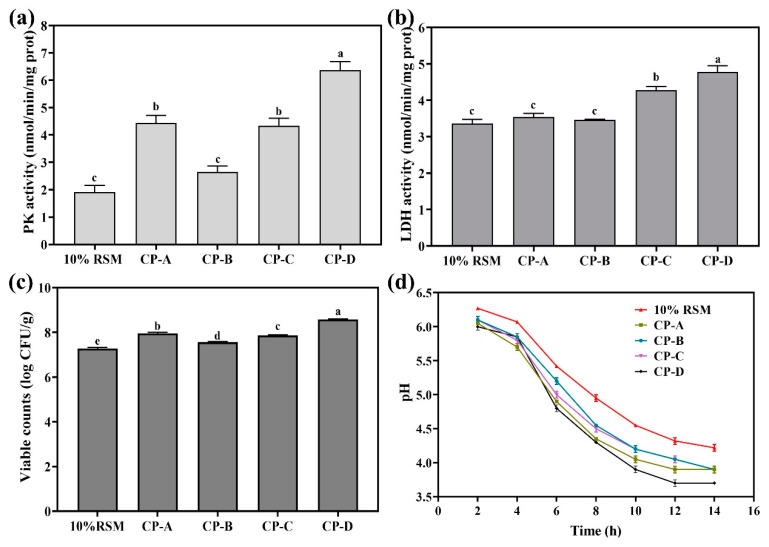
Effect of 10% (*w*/*v*) RSM-based composite protectants on the enzymes PK and LDH, viable counts, and fermented performance of *Lb. acidophilus* NCFM after spray drying. Images (**a**–**d**) show PK activity, LDH activity, viable counts, and fermented performance of strain NCFM from 10% (*w*/*v*) RSM-based composite protectants, respectively. Different letters from “a to e” on the top of each column indicate significant differences (*p* < 0.05).

**Table 1 microorganisms-12-01094-t001:** Effect of spray drying on the DPH (1,6-diphenyl-1,3,5-hexatriene) fluorescence polarization (*p*) and micro-viscosity (*η*) values of the selected 3 LAB strains’ membrane.

Treatment	*Lb. acidophilus* NCFM		*Lb*. *bulgaricus* CICC 6097		*Lpb. plantarum* CICC 21839	
	*p* Value	*η*	*p* Value	*η*	*p* Value	*η*
Control group	0.295 ± 0.002 ^a^	3.579 ± 0.4 ^a^	0.290 ± 0.002 ^a^	3.419 ± 0.2 ^a^	0.289 ± 0.003 ^a^	3.392 ± 0.12 ^a^
1% RSM group	0.320 ± 0.003 ^b^	4.451 ± 0.12 ^b^	0.318 ± 0.003 ^b^	4.491 ± 0.38 ^b^	0.362 ± 0.003 ^c^	7.352 ± 0.03 ^c^
10% RSM group	0.318 ± 0.009 ^ab^	4.311 ± 0.33 ^ab^	0.309 ± 0.005 ^b^	4.093 ± 0.19 ^ab^	0.303 ± 0.005 ^b^	3.870 ± 0.21 ^b^

The results are presented as means ± standard deviations. Means with different letters in a column are significantly different (*p* < 0.05).

**Table 2 microorganisms-12-01094-t002:** *Lb. acidophilus* NCFM KEGG pathways analysis of differentially expressed genes.

Sample Group		Pathways	Related Enzymes (Genes)
1% RSM group vs. control group	Up-regulated	Ribosome;	50S ribosomal protein (L34, L28, L30, L35, S21)/30S ribosomal protein (S14, L31, L33)
		ABC transporters	ABC transporter ATP-binding protein (*MdlB*)/energy-coupling factor transporter ATPase/ABC transporter permease
	Down-regulated	Phosphotransferase system (PTS)	PTS sugar transporter subunit IIA IIB IIC (*crr*, *celA*, *celB*)/PTS beta-glucoside transporter subunit BglF (*scrA*)
		Starch and sucrose metabolism	sucrose-6-phosphate hydrolase (*scrB*)/β-phosphoglucomutase (β-PGM) (*pgmB*)
		Glycolysis	pyruvate synthase (*pyk*)
		Cysteine metabolism	cystathionine beta-lyase (*metC*)
		Pyruvate metabolism	fumarate reductase flavoprotein subunit (*FrdA*)/class II fumarate hydratase (*fumC*)/NAD-dependent malic enzyme (*maeA*)/L-lactate dehydrogenase (*ldh*);
		Glycerolipid metabolism	dihydroxyacetone kinase subunit L (*dhal*, *dhaK*)/glycerol kinase GlpK (*glpK*)
10% RSM group vs. 1% RSM group	Up-regulated	Pyrimidine metabolism	ribonucleoside-triphosphate reductase (*nrdJ*)/thioredoxin-disulfide reductase (*trxB*)
		ABC transporters	glycerol-3-phosphate ABC transporter ATP-binding protein (*ugpC*)
		Phosphotransferase system (PTS)	PTS lactose/cellobiose transporter subunit IIA (*celC*)/PTS sugar transporter subunit IIB, IIC (*celA*, *celB*)
		Thiamine metabolism	phosphomethylpyrimidine kinase (*thiD*)
		Pyruvate metabolism	acetate kinase (*ackA*)
		Amino sugar and nucleotide sugar metabolism	glucosamine-6-phosphate deaminase (*nagB*)/nicotinic acid mononucleotide NaMN (*nadD*)

## Data Availability

The original contributions presented in the study are included in the article, further inquiries can be directed to the corresponding author.

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
