# Peer review of "Screening the Protective Agents Able to Improve the Survival of Lactic Acid Bacteria Strains Subjected to Spray Drying Using Several Key Enzymes Responsible for Carbohydrate Utilization"

_microorganisms, 2024, doi:10.3390/microorganisms12061094_

Round 1

Reviewer 1 Report

Comments and Suggestions for Authors

1.- In lines 64 and 65 the authors mention that spray drying is an alternative to freeze-drying, this is not correct, they are very different processes. The authors should correct this sentence.

 2.- In the materials and methods section when centrifugation is performed in terms of g, the authors should describe the characteristics of the rotor and centrifuge used. Authors should add this information in all cases.

 3.-Authors should justify why they use pH as acid production, instead of calculating lactic acid production.

 4.-The authors point out that they selected strain NCFM for the transcriptome studies because this strain showed the least resistance to the spray drying process; however, it would be interesting for the authors to also take the strain that showed the highest resistance to spray drying to achieve a better comparison and to be able to better support their conclusions. Therefore, the authors should justify why they did not investigate the strain with the highest resistance to the spray-drying process.

Author Response

Responds to the referee 1’s comments: green marks

    We sincerely appreciate your valuable comments which are helpful for improving the quality of our manuscript. Corrections have been made point to point, and with green marks in the revised manuscript. We would like to wish these corrections are able to meet with approval.

1. Response to in lines 64 and 65 the authors mention that spray drying is an alternative to freeze-drying, this is not correct, they are very different processes. The authors should correct this sentence.

Answer: We thank for this good comment. We have redrafted this sentence (see line 61-62).

Revised version: Commonly, spray drying is a cost-effective method that produces active probiotic powder compared to freeze drying [4].

2. Response to in the materials and methods section when centrifugation is performed in terms of g, the authors should describe the characteristics of the rotor and centrifuge used. Authors should add this information in all cases.

Answer: Thanks for your kind guidance. In our experiments, the centrifuge’s rotor radius is 15 cm and the model of centrifuge is GL-20G-II. We have added the characteristics of centrifuge in line 137. At the same time, we have converted centrifugal force “g” uniformly into rotational speed “rpm/min”. Please check lines 138, 170, 264, 353 and 359.

3. Response to authors should justify why they use pH as acid production, instead of calculating lactic acid production.

Answer: We really thank your terrific comment. Firstly, it is reported that the pH curve can reflect the fermentation capacity of lactic acid bacteria (LAB) because pH value represents the H+ concentration in the fermentation medium (Ji et al., 2023). Thus, we measured the fermentation activity of LAB strains by the change in pH due to lactic acid production. Secondly, Ghandi et al. used pH values to measure fermentation activity of spray-dried Lactococcus lactis and obtained the same results (Ghandi et al., 2013). Finally, we have added the cause of using pH as acid production in section 3.1.2. Please check line 386-388.

(References: Ji G, Liu G, Li B, Tan H, Zheng R, Sun X, He F. Influence on the aroma substances and functional ingredients of apple juice by lactic acid bacteria fermentation[J]. Food Bioscience. 2023, 51: 102337. https://doi.org/10.1016/j.fbio.2022.102337

Ghandi, A., Powell, I. B., Broome, M., & Adhikari, B. (2013). Survival, fermentation activity and storage stability of spray dried Lactococcus lactis produced via different atomization regimes. Journal of Food Engineering, 115(1), 83-90. https://doi.org/10.1016/j.jfoodeng.2012.09.022 )

Revised version: pH changes of milk fermented by the tested LAB strains were used to indicate the fermentation activity of these microorganisms before and after spray drying, because generally pH value represents the H+ concentration in the fermentation medium [26].

4. Response to the authors point out that they selected strain NCFM for the transcriptome studies because this strain showed the least resistance to the spray drying process; however, it would be interesting for the authors to also take the strain that showed the highest resistance to spray drying to achieve a better comparison and to be able to better support their conclusions. Therefore, the authors should justify why they did not investigate the strain with the highest resistance to the spray-drying process.

Answer: We thank for this good comment. Firstly, we found that the NCFM strain was more sensitive to spray drying than strains Lb. bulgaricus CICC 6097 and Lpb. plantarum CICC 21839 based on the results of survival and key enzyme activities. Thus, strain NCFM cells were used as a case study, and we expect to further analyze the reasons for the decrease of lactic acid bacteria (LAB) activity caused by spray drying with transcriptome data. Secondly, we referred to the research method of Corcoran et al. (2004), who reported that Lactobacillus rhamnosus GG had the poor heat resistance during spray drying compared to two other strains (Lactobacillus rhamnosus E800 and Lactobacillus salivarius UCC 500). At the same time, they also took Lactobacillus rhamnosus GG as an example, and their results showed that the heat resistance of the Lactobacillus rhamnosus GG strain was improved at the protection of skim milk. Thirdly, our work indicated measuring enzyme activity related to glycolysis process could be used as a promising tool to screen the effective spray-dried protective agent for LAB cells. The four enzyme activities (HK, β-GAL, PK and LDH) of spray-dried strain NCFM were significantly lower than those of the other two LAB cells. So, we expected that the cells which had worse tolerance to spray drying would be used as a case to illustrate our objectives. Finally, we have revised the description in line 520-522 and added a new reference to favor our opinions.

(References: Corcoran, B. M., Ross, R. P., Fitzgerald, G. F., & Stanton, C. Comparative survival of probiotic lactobacilli spray‐dried in the presence of prebiotic substances. Journal of Applied Microbiology 2004, 96(5), 1024-1039. https://doi.org/10.1111/j.1365-2672.2004.02219.x )

Revised version: Comprehensively, we picked up strain NCFM as the tested strain to further investigate the damaged mechanism of spray drying towards cells using transcriptomics method referring to the study of Corcoran et al. (2004) [30].

In a word, we heartily thank the reviewer for all comments and suggestions.

Reviewer 2 Report

Comments and Suggestions for Authors

The study screened the desirable protectants available for improving the survival of LAB by measuring their enzymatic activity. They found that measuring the key enzyme LDH and PK could be used as a promising tool to screen the potential protective agent for improving the tolerance of LAB strains against spray drying. They constructed the convenient, quick and effective for evaluating the LAB activity. The topic is interesting and meaningful for industrial application of LABs.

1.        L 106 delete “about”

2.        L115 rephrased

3.        L113-117 the hypothesis and objective should be presented in this paragraph

4.        L121-130 delete and combine this information of reagents in the methods

5.        L151 the digit should not start a sentence, please check the whole text.

6.        Please provided the reference for determination of β-galactosidase, hexokinase, pyruvate kinase, lactate dehydrogenase

7.        The full name appears only when first appearing, please the whole text.  

8.        How to analyze the data? Model? Fixed effect?

9.        The title of Figure 1 is not related with contents

Comments on the Quality of English Language

the text should be improved by  Einglish native speaker.

Author Response

Responds to the referee2’s comments: red marks

    We are grateful for your valuable comments which are very beneficial for revising our manuscript. We have made modifications after reading your comments carefully. The revised portions are marked with red in the manuscript. We would like to wish these corrections are able to meet with approval.

1.Response to line 106 delete “about”.

Answer: We thank for this valuable advice to us. We have deleted “about”. Please check line 100-101.

Revised version: …up-regulating LDH enzyme genes associated with pyruvate metabolism and membrane fatty acids metabolic pathways [18].

2. Response to line 115 rephrased.

Answer: We really thank your terrific comment. We have revised the description between line 113 and 115.

Revised version: Lastly, the study utilized the enzymes LDH and PK as pivotal indicators to screen the desirable protective agents which improved the resistance of selected LAB strains to spray drying.

3. Response to line 113-117 the hypothesis and objective should be presented in this paragraph.

Answer: We thank for this valuable advice to us. We have redrafted the description between line 108 and 115.

Revised version: Therefore, the objective of this study was designed to look for appropriate thermo-protective agents for selected LAB cells exposed to spray drying, with a focus on measuring the activity of carbohydrate utilization-related enzymes. Firstly, the study examined the detrimental effects of spray drying on the cell membranes of the LAB strains and their enzyme activity. Subsequently, it delved into understanding how spray drying produces the injury to LAB cells through transcriptome data analysis. Lastly, the study utilized the enzymes LDH and PK as pivotal indicators to screen the desirable protective agents which improved the resistance of selected LAB strains to spray drying.

4. Response to line 121-130 delete and combine this information of reagents in the methods.

Answer: We thank for this comment. We have deleted and combined this information of reagents in the methods. Please see line 118-124.

Revised version: Recover skim milk (RSM) was purchased from Fonterra group Co., Ltd., Shanghai, China. One step bacterial active protein extraction kit, Bicinchoninic Acid (BCA) protein assay kit, LDH assay kit, PK assay kit, β-GAL assay kit, and HK assay kit were bought from Suzhou Comin Biotechnology Co., Ltd., Jiangsu, China. Live/Dead bacLight bacterial viability kit L7012, sodium glutamate, trehalose, and sucrose were all prepared by Life technology Co., Ltd. (Tianjin, China) All other chemicals were analytical grade reagents and were prepared by Solarbio Co., Ltd. (Beijing, China).

5. Response to line 151 the digit should not start a sentence, please check the whole text.

Answer: We thank you for this query. We have revised the description which the digit was used as the beginning sentences in the whole text. Please check line 144, 145, 169, 263, 264-265, 340, 346, 352, and 358.   

Revised version: The 1% (w/v) or 10% (w/v) RSM medium (Line 144). / The suspension of each LAB strain (0.5 mL) (Line 145). / Spray-dried powders of the selected 3 LAB strains with protectants (0.5 g) (Line 169). / spray-dried powder of each strain (0.5 g) (Line 263). / The 1,6-Diphenyl-1,3,5-hexatriene (DPH, 2 mL, 2 μmol/L) (Line 264-265). / The 5 mL suspension of strain NCFM (Line 340). / The 10% (w/v) RSM spray-dried powders (Line 346). / Approximate 0.5g spray-dried powders (Line 352). / Approximate 0.5 g spray-dried powders (Line 358).

6. Response to please provided the reference for determination of β-galactosidase, hexokinase, pyruvate kinase, lactate dehydrogenase.

Answer: We are grateful for your good comments. We have added relevant references about determination of β-galactosidase, hexokinase, pyruvate kinase, lactate dehydrogenase in section 2.6. Please check line 187-188, 208, 225-226, and 242.

Revised version:

Line 187-188: HK activity of each encapsulated strain was determined with HK assay kit based on the report of Zhou et al. (2018) [19].

Line 208: β-GAL activity of each encapsulated strain was determined with β-GAL assay kit [12].  

Line 225-226: PK activity of each encapsulated strain was determined with PK assay kit as described by Li et al. (2016) [20].

Line 242: LDH activity of each encapsulated strain was determined with LDH assay kit [19].

7. Response to the full name appears only when first appearing, please the whole text.

Answer: We are grateful for your good comments. We have revised the full names and their abbreviations in the whole text. Please check line 119-120, 186, 207, 224 and 241.

Revised version:

Line 119-120: LDH assay kit, PK assay kit, β-GAL assay kit, and HK assay kit

Line 186/207/224/241: 2.6.2 Determination of HK activity/ 2.6.3 Determination of β-GAL activity/ 2.6.4 Determination of PK activity/ 2.6.5 Determination of LDH activity.

8. Response to how to analyze the data? Model? Fixed effect?

Answer: We thank you for this query. We are sorry not to make a clear description to this point in our manuscript. We have redrafted this description in the section. Please check line 363-366.

Revised version: All experiments were at least repeated three times. The results were processed by Origin 9.1 and GraphPad Prism 9.0. The experiment was expressed in the form of mean ± standard deviation (SD). Significance differences among treatments at p < 0.05 among results were compared by the Duncan’s new multiple range test of one-way (ANOVA) test by SPSS 22.0 statistical software.

9. Response to the title of Figure 1 is not related with contents.

Answer: We are grateful for your good comments. We have revised the title of Figure 1. Please check line 397-399.

Revised version: Figure 1. The images of (a), (b) and (c) are pH changes in milk fermented by Lb. acidophilus NCFM, Lb. bulgaricus CICC 6097 and Lpb. plantarum CICC 21839 after spray drying, respectively.

10. Response to the text should be improved by English native speaker.

Answer: We are grateful for your good comments. We have the native English expert checked the language again after the revision of the manuscript. And we have added information about the English expert in section Acknowledgments (see line 785-786).

Revised version: Authors appreciate Dr. Xiaoqi Sun who works in the Department of Surgery, University of South Florida, USA, for polishing language and modifying paper.

    We sincerely thank the reviewer to point out the problems for improving the quality of our manuscript.

Reviewer 3 Report

Comments and Suggestions for Authors

A very interesting manuscript on the effect of various protective agents on the survival of specific LAB strains subjected to spay drying. The study is well designed and the manuscript is very informative; however, the following should be addressed before publication:

1.       Please pay attention to the use of italics, the use of correct species names and the use of the updated LAB nomenclature (l. 21, 74, 134, 704, 707, 708)

2.       L. 171. The authors incubated for 48 h or until colony formation? due to the stress of spray drying, the incubation may take a little bit longer. Have the authors considered the possibility of prolonged incubation?  

3.       L. 174-175. Would this determination be performed in the original data or in the log-transformed ones? According to l. 148 & 152, a total population of ~5 x 10^9 CFU was spray dried. According to Table 1, the viable count of strain NCFM (1% RSM) was 6.8, which is ~6.3 x 10^6. Survival by using log transformed values is 65.7%, by using the original data is 0.1%. Which one is correct? Please use statistical and biological terms and not justify this selection because of its presence in the literature, both calculations have been published. Please apply the changes throughout the text, where applicable.

4.       L. 197 and throughout the text. How is the control group treated? Please be specific.

5.       L. 201-203, 239-240 & 257-258 present different centrifugation conditions than 220-221. Why?

6.       Please rephrase ‘3 seconds at 10 intervals for 30 repetitions’ to become scientifically relevant and comprehensible.

7.       L. 314, this paragraph is not about genome.

8.       Important information is missing from paragraph 2.9, which are present in l. 547-549. Please describe in detail how the control and the sample conditions were treated in order to facilitate understanding of this experiment.

Comments on the Quality of English Language

The use of English language is poor. Although it can be improved by the editorial team, the sentences in lines 29-30, 59-61, 105-108, 110-111, 299, 757-759,  need to be rephrased by the authors to improve scientific accuracy

Author Response

Responds to the referee3’s comments: blue marks

    We thank for your time to review our manuscript and give us comments. We have made modifications after reading your comments carefully. Corrections have been made point to point, and with blue marks in the revised manuscript.

1. Response to please pay attention to the use of italics, the use of correct species names and the use of the updated LAB nomenclature (Line 21, 74, 134, 704, 707, 708)

Answer: We thank you for your kindness and patience to point out the errors in our manuscript. We have corrected the species names of the whole manuscript. Please check line 24-25, 70, 128-129, 683, and 686.

Revised version: Lactobacillus delbrueckii subsp. bulgaricus CICC 6097, Lactiplantibacillus plantarum CICC 21839, Lactobacillus acidophilus NCFM (Line 24-25); Lacticaseibacillus casei BL23 (Line 70); Lactobacillus deribreckkii subsp. bulgaricus CICC 6097 (Lb. bulgaricus CICC 6097) and Lactiplantibacillus plantarum CICC 21839 (Lpb. plantarum CICC 21839) (Line 128-129); Lacticaseibacillus rhamnosus GG (Line 683); Lactiplantibacillus plantarum L1 and Limosilactobacillus fermentum L2 (Line 686)

2. Response to line 171. The authors incubated for 48 h or until colony formation? due to the stress of spray drying, the incubation may take a little bit longer. Have the authors considered the possibility of prolonged incubation?

Answer: We thank for this query. We have revised the description in line 162-163.

Revised version: The 1.0 mL of LAB strains were plated into de Man-Rogosa-Sharpe medium (MRS) agar medium and incubated at 37°C until colonies formed.

3. Response to Line 174-175. Would this determination be performed in the original data or in the log-transformed ones? According to line 148 & 152, a total population of ~5 x 109 CFU was spray dried. According to Table 1, the viable count of strain NCFM (1% RSM) was 6.8, which is ~6.3 x 106. Survival by using log transformed values is 65.7%, by using the original data is 0.1%. Which one is correct? Please use statistical and biological terms and not justify this selection because of its presence in the literature, both calculations have been published. Please apply the changes throughout the text, where applicable.

Answer: We thank for this query. We are very sorry not to make a clear description. We performed in log-transformed data to calculate the survival percentage of 3 selected LAB strains. And we have revised the description in line 163-167 and Table 1. At the same time, Yin et al. also used log-transformed data to calculate survival of spray-dried Lactobacillus rhamnosus GG microcapsules.  

(References: Yin, M., Yuan, Y., Chen, M., Liu, F., Saqib, M. N., Chiou, B.-S., & Zhong, F. (2022). The dual effect of shellac on survival of spray-dried Lactobacillus rhamnosus GG microcapsules. Food Chemistry, 389, 132999. https://doi.org/10.1016/j.foodchem.2022.132999)

4. Response to line 197 and throughout the text. How is the control group treated? Please be specific.

Answer: We thank for this query. We are very sorry not to make a clear description. We have revised this description in line 188-192, 208-209, 226 and 242-243.

Revised version:

Line 188-192: The LAB strains without experiencing spray drying were used as the control group, and each strain was inoculated into 5 mL MRS broth at 5% (w/v) inoculum size, and then anaerobically incubated at 37°C for 18 h. After that, cells were collected by centrifugation (5000 rpm/min, 4°C, and 10 min), and the collected cells were re-hydrated in 5 mL sterile water for further analysis.

Line 208-209, 226 and 242-243: The control group was prepared as described in section 2.6.2.

5. Response to line 201-203, 239-240 & 257-258 present different centrifugation conditions than 220-221. Why?

Answer: We thank for this query. We prepared the samples according to the related enzyme assay kits’ instructions, and we referred to the centrifugation conditions of the enzyme assay kits. Moreover, Zhou et al. and Cheng et al. also both used the same test conditions. At the same time, we added the relevant references. Please check line 187-188, 208, 225-226 and 242.

(References: Cheng, Z., Yan, X., Wu, J., Weng, P., & Wu, Z. Effects of freeze drying in complex lyoprotectants on the survival, and membrane fatty acid composition of Lactobacillus plantarum L1 and Lactobacillus fermentum L2. Cryobiology 2022, 105, 1-9. https://doi.org/10.1016/j.cryobiol.2022.01.003

Zhou, F., Jiang, X., Wang, T., Zhang, B., & Zhao, H. Lyciumbarbarum polysaccharide (LBP): A novel prebiotics Candidate for Bifidobacterium and Lactobacillus. Frontiers in Microbiology 2018, 9, 1034. https://doi.org/10.3389/fmicb.2018.01034)

Revised version:

Line 187-188: HK activity of each encapsulated strain was determined with HK assay kit based on the report of Zhou et al. (2018) [19].

Line 208: β-GAL activity of each encapsulated strain was determined with β-GAL assay kit [12]. 

Line 225-226: PK activity of each encapsulated strain was determined with PK assay kit as described by Li et al. (2016) [20].

Line 242: LDH activity of each encapsulated strain was determined with LDH assay kit [19].

6. Response to please rephrase ‘3 seconds at 10 intervals for 30 repetitions’ to become scientifically relevant and comprehensible.

Answer: We thank for this valuable advice to us. We are very sorry not to make a clear description. We have revised the description about ultrasound treatment conditions. Please check line 194-196, 211-213, 229-230, 245-247.

Revised version:

Line 194-196: The conditions performed for the cell suspension were sonicated for 3 seconds, and then paused for 10 seconds. The ultrasound treatment for each sample was repeated 30 times.

Line 211-213, 229-230 and 245-247: The cell suspension was sonicated for 3 seconds and then paused for 10 seconds. The ultrasound treatment for each sample was repeated 30 times.

7. Response to line 314, this paragraph is not about genome.

Answer: We really thank your terrific comment. We are very sorry for this to occur. We have revised the description between line 302 and 303.

Revised version: The transcriptome sequencing of strain NCFM was determined by referring to Yang et al. (2021) method with some modifications [23].

8. Response to important information is missing from paragraph 2.9, which are present in line 547-549. Please describe in detail how the control and the sample conditions were treated in order to facilitate understanding of this experiment.

Answer: We really thank your terrific comment. We have revised the description about preparing samples in section 2.9. Please check line 302-315.

Revised version: The transcriptome sequencing of strain NCFM was determined by referring to Yang et al. (2021) method with some modifications [23]. Firstly, the activated strain NCFM cells were inoculated into 1000 mL MRS broth at 5% (w/v) inoculum volume. The cells collected by centrifugation of 5000 rpm/min at 4°C for 10 min were anaerobically incubated at 37°C for 18 h. Ten grams of wet bacterial pellets were segmented into two equal parts. One part (5 g) was used to prepare the control sample, and another part (5 g) was used to prepare spray dried sample. Secondly, spray-dried powder of strain NCFM (5 g) was dissolved in 50 mL 0.01 M PBS (pH 7.0), and then centrifugated by 5000 rpm/min at 4°C for 10 min to collected cells. The cells were added into 100 mL MRS medium for incubation at 37°C for 18 h. Then the cultivated medium was centrifugated with 5000 rpm/min at 4°C for 10 min to collect cell pellets for RNA extraction. At the same time, 5 g strain NCFM without spray drying treatment as the control group was inoculated into 100 mL MRS medium incubated at 37°C for 18 h. Then the incubated medium of the control group was centrifugated with 5000 rpm/min at 4°C for 10 min to collect cell pellets for RNA extraction.

9. Response to the use of English language is poor. Although it can be improved by the editorial team, the sentences in lines 29-30, 59-61, 105-108, 110-111, 299, 757-759, need to be rephrased by the authors to improve scientific accuracy.

Answer: We thank for this nice comment. We have the native English expert checked the language again after the revision of the manuscript. And we have added information about the English expert in section Acknowledgments (see line 785-786). At the same time, we revised the description in line 31-33, 56-59, 99-105, 287-289 and 733-735.

Revised version:

Line 31-33: Next, transcriptome data of Lb. acidophilus NCFM showed that 10% (w/v) RSM improved the down-regulated expressions of genes encoding PK (pyk) and LDH (ldh) after spray drying compared to 1% (w/v) RSM.

Line 56-59: The International Dairy Federation recommends that the desirable probiotic products conferring beneficial effects on the host should contain at least the live counts of LAB which are more than 106 CFU/mL [2].

Line 99-105: For instance, it was reported that Lactiplantibacillus plantarum LIP-1 can withstand the stress of a long-term acid environment by up-regulating LDH enzyme genes associated with pyruvate metabolism and membrane fatty acids metabolic pathways [18]. Furthermore, in a study by Zhou et al. (2018), the addition of 5% Lycium barbarum polysaccharides as protective agent not only enhanced the viability of Bifidobacterium Bb-02 but also improved the activity of β-GAL, LDH, HK and PK of bacterial cells exposed to freeze drying by overexpressing the genes of transmembrane transport system (opp) and glycolysis process [19].

Line 287-289: The stained bacterial suspension (5 μL) was placed on slide to obtain the micro-images using fluorescence microscope (DM500, Leica Co., Ltd., Wetzlar, Germany) [21].  

Line 733-735: The oxidation-reduction process and expression genes responsible for the sucrose (scrB) and trehalose (pgmB) utilization of strain NCFM were significantly down-regulated after this bacterium experienced the stress of spray drying (Table 3 and Fig 5A).

Line 785-786: Authors appreciate Dr. Xiaoqi Sun who works in the Department of Surgery, University of South Florida, USA, for polishing language and modifying paper.

    We sincerely thank the reviewer to point out the problems for improving the quality of our manuscript.

Round 2

Reviewer 3 Report

Comments and Suggestions for Authors

I am afraid I will have to insist on two comments:

l. 162-163. In the original manuscript the authors mentioned that incubation took place for 48 h. The reviewer commented that due to the stress imposed by spray drying, 48 h may not be enough and prolonged incubation may have been necessary (this is something that should have been assessed during pre-experiments). The authors changed ’48 h’ to ‘until colony formation’. The manuscript should contain the exact incubation conditions used, which are in accordance with the results presented. The ‘until colony formation’ how many days include?

L. 174-175. This type of data should not be log-transformed before statistical analysis. The authors claim that they log-transformed the data and then applied statistical analysis because Yin et al. (2022) (https://doi.org/10.1016/j.foodchem.2022.132999) also applied the same strategy. Please read Table 1, in the last column where Lin et al. (2022) report the survival rate, they did not use the log-transformed data. In addition, Corcoran et al. (2004) (https://doi.org/10.1111/j.1365-2672.2004.02219.x) also calculated the survival on the original data and not the log-transformed, because this is the correct approach (there are hundreds of manuscripts applying the same). Surely the opposite may also be found in the literature, but this is not correct.

In addition:

Paragraph 2.9. The authors mentioned that ‘One part (5 g) was used to prepare the control sample, and another part (5 g) was used to prepare spray dried sample.’ What happened to the part consisting the control sample while the other part was spray dried and prepared for the final incubation before RNA extraction? The authors should report that. Then, the authors reported that they incubated both parts at 37 oC for 18 h before RNA extraction. Isn’t 18 h incubation at optimum temperature enough time to repair any problems that spray-drying or ‘what happened to the control sample’ may have caused? Are the authors certain that the differences they observed after RNAseq can be attributed to the spray-drying alone and not to the combination of spray-drying and ‘what happened to the control sample while was spray dried and prepared’ or to other factors (including technical issues)? Discussing the capabilities of the methods used, in biological terms, is a vital part of the discussion section and is the only way to advance science; please add such a paragraph in the discussion.

Comments on the Quality of English Language

Minor issues still persist, the editorial team of MDPI will polish them

Author Response

Responds to the referee 1’s comments: red and yellow marks

We sincerely appreciate your valuable comments which are helpful for improving the quality of our manuscript. Corrections have been made point to point, and with red and yellow marks in the revised manuscript. We would like to wish these corrections are able to meet with approval.

1. Response to Line 162-163. In the original manuscript the authors mentioned that incubation took place for 48 h. The reviewer commented that due to. The authors changed “48 h” to “until colony formation”. The manuscript should contain the exact incubation conditions used, which are in accordance with the results presented. The ‘until colony formation’ how many days include?

Answer: We thank for the critical comments. We are sorry not to make a clear description to this point in our manuscript due to we have misunderstood the reviewer's first comments. Indeed, you are right, and the stress imposed by spray drying may not be enough for incubation to observe colony formation. In our experiments, the colonies of our LAB strains after incubated at 37°C for 48 h could be observed. That is why we counted the live cells after 48 h incubation. Thus, we have revised this description in line 163-164 according to your comments. At the same time, Gong et al. (2019) also cultured LAB plates at 37°C for 48 h for the numeration of live cells.

(References: Gong, P., Di, W., Yi, H., Sun, J., Zhang, L., & Han, X. (2019). Improved viability of spray-dried Lactobacillus bulgaricus sp1.1 embedded in acidic-basic proteins treated with transglutaminase. Food Chemistry, 281, 204-212. https://doi.org/10.1016/j.foodchem.2018.12.095)

Revised version: The plates were incubated at 37°C for 48 h to count viable cells.

2. Response to Line 174-175. This type of data should not be log-transformed before statistical analysis. The authors claim that they log-transformed the data and then applied statistical analysis because Yin et al. (2022) (https://doi.org/10.1016/j.foodchem.2022.132999) also applied the same strategy. Please read Table 1, in the last column where Lin et al. (2022) reported the survival rate, they did not use the log-transformed data. In addition, Corcoran et al. (2004) (https://doi.org/10.1111/j.1365-2672.2004.02219.x) also calculated the survival on the original data and not the log-transformed, because this is the correct approach (there are hundreds of manuscripts applying the same). Surely the opposite may also be found in the literature, but this is not correct.

Answer: We really appreciate you for the critical comments. Your comments helped us improve the preciseness in method and data description. Based on your comments, we have deleted the expression “survival rate” of LAB strains from our manuscription, and only used the live cell counts to illustrate the viability of LAB strains treated before and after spray drying (see line 161-165). Thus, we changed the data of table 1 (viable counts of three LAB strains) into figure 1 for clearer reading. Please check Figure1, and all corrections in the corresponding places of this revised manuscript have been made and marked with yellow.

Revised version: The 0.5 g spray-dried powder of 3 selected LAB strains with protectants, was dissolved in 5 mL sterile water to release cells. The cells were added into 10 mL sterile water to make a series of dilutions. Then, the 1.0 mL diluted suspensions were plated into MRS agar medium. The plates were incubated at 37°C for 48 h to count viable cells. The viable counts of cells were presented as log CFU per gram of dried powders (log CFU/g) [4].

3. Response to Paragraph 2.9. The authors mentioned that ‘One part (5 g) was used to prepare the control sample, and another part (5 g) was used to prepare spray dried sample.’ What happened to the part consisting the control sample while the other part was spray dried and prepared for the final incubation before RNA extraction? The authors should report that. Then, the authors reported that they incubated both parts at 37 ℃ for 18 h before RNA extraction. Isn’t 18 h incubation at optimum temperature enough time to repair any problems that spray-drying or ‘what happened to the control sample’ may have caused? Are the authors certain that the differences they observed after RNAseq can be attributed to the spray-drying alone and not to the combination of spray-drying and ‘what happened to the control sample while was spray dried and prepared’ or to other factors (including technical issues)? Discussing the capabilities of the methods used, in biological terms, is a vital part of the discussion section and is the only way to advance science; please add such a paragraph in the discussion.

Answer: We really thank you to point out the problem in our manuscript, and we really feel sorry not to make a right description in making samples before RNA extraction. Your comments helped us improve the preciseness in method description. Actually, the method for RNA extraction was prepared by referring to Zhang et al. (2020) approach with some modifications. Firstly, the activated strain NCFM cells were anaerobically incubated at 37°C for 18 h for the preparation of spray dried and control samples. Then the collected cells were separated two parts, one used as spray dried samples and the other used as control samples, which were immediately stored in liquid nitrogen for RNA extraction. In other words, the RNA of control and treatment groups was measured under the same conditions, and no changes might take place. Based on your comments, we have made a correction to this point in line 301-309, and relevant reference was added (see line 868-870).

Revised version: Firstly, the activated strain NCFM cells were inoculated into 1000 mL MRS broth at 5% (w/v) inoculum volume and were anaerobically incubated at 37°C for 18 h. The cells were collected by centrifugation with 5000 rpm/min at 4°C for 10 min. Ten grams of wet bacterial pellets were segmented into two parts. One part (5 g) of strain NCFM, after mixed with 1% or 10% RSM, was spray-dried and immediately dissolved into 50 mL 0.01 M PBS (pH 7.0) to collect cell pellets by centrifugation with 5000 rpm/min at 4°C for 10 min. Another part (5 g) of strain NCFM cells without spray drying was used as the control group. And the resulting pellets from the control groups and spray dried groups were immediately stored in liquid nitrogen for RNA extraction. (References:[23] Zhang C, Gui Y, Chen X, et al. Transcriptional homogenization of Lactobacillus rhamnosus hsryfm 1301 under heat stress and oxidative stress[J]. Applied microbiology and biotechnology 2020, 104(6): 2611-2621. https://doi.org/10.1007/s00253-020-10407-3 )

In a word, we heartily thank the reviewer for all comments and suggestions.
